# An open-source platform for head-fixed operant and consummatory behavior

Adam Gordon-Fennell[1], Joumana M Barbakh[1], MacKenzie T Utley[1], Shreya Singh[1], Paula Bazzino[2,3], Raajaram Gowrishankar[1], Michael R Bruchas[1], Mitchell F Roitman[2,3], Garret D Stuber[1]*

[1]Center for the Neurobiology of Addiction, Pain, and Emotion, Department of Anesthesiology and Pain Medicine, Department of Pharmacology, University of Washington, Seattle, United States; [2]Department of Psychology, University of Illinois at Chicago, Chicago, United States; [3]Graduate Program in Neuroscience, University of Illinois at Chicago, Chicago, United States

**Abstract** Head-fixed behavioral experiments in rodents permit unparalleled experimental control, precise measurement of behavior, and concurrent modulation and measurement of neural activity. Here, we present OHRBETS (Open-Source Head-fixed Rodent Behavioral Experimental Training System; pronounced 'Orbitz'), a low-cost, open-source platform of hardware and software to flexibly pursue the neural basis of a variety of motivated behaviors. Head-fixed mice tested with OHRBETS displayed operant conditioning for caloric reward that replicates core behavioral phenotypes observed during freely moving conditions. OHRBETS also permits optogenetic intra-cranial self-stimulation under positive or negative operant conditioning procedures and real-time place preference behavior, like that observed in freely moving assays. In a multi-spout brief-access consumption task, mice displayed licking as a function of concentration of sucrose, quinine, and sodium chloride, with licking modulated by homeostatic or circadian influences. Finally, to highlight the functionality of OHRBETS, we measured mesolimbic dopamine signals during the multi-spout brief-access task that display strong correlations with relative solution value and magnitude of consumption. All designs, programs, and instructions are provided freely online. This customizable platform enables replicable operant and consummatory behaviors and can be incorporated with methods to perturb and record neural dynamics in vivo.

*For correspondence:
gstuber@uw.edu

Competing interest: The authors declare that no competing interests exist.

## Editor's evaluation

This important study by Gordon-Fennell et al. presents a low-cost, open-source platform for measuring action elicitation and consummatory behavior in head-fixed animals. The authors present exceptional evidence that this platform allows animals to perform a truly voluntary action whilst their head is held still. The results have the potential to have a broad impact in the field as many labs start to move towards measuring head-fixed behavior effectively, although this is said with the caveat that such behavior will never be an ideal replication of naturalistic behavior.

## Introduction

Studying mouse behavior under head-fixed conditions offers many distinct advantages over freely moving conditions. At the cost of reduced naturalistic behavior and enhanced stress, head fixation offers high degrees of behavioral control that enables consistent delivery of stimuli to the animal, precise measurement of behavior, and isolation of subcomponents of behavior (*Bjerre and Palmer, 2020*). Holding the mouse stable permits a wide range of behavioral experiments that have features

that are challenging or impossible to conduct reliably in the freely moving condition including the delivery of somatosensory stimuli to select locations, temporally precise odor delivery (*Han et al., 2018*), presentation of visual stimuli to fixed parts of the visual field (*Krauzlis et al., 2020*; *Aguillon-Rodriguez et al., 2021*), temperature manipulations (*Jung et al., 2022*), and high-resolution video recording of facial expression or paw movement (*Dolensek et al., 2020*; *Mathis et al., 2018*). Eliminating or controlling physical approach behaviors, but not all movement (*Hughes et al., 2020*), allows for greater isolation of both appetitive and consummatory behaviors and related neuronal dynamics. By removing turning associated with locomotion, head-fixed behavioral approaches offer enhanced compatibility with neuroscience approaches that require tethers including optogenetics and fiber photometry. Furthermore, head fixation is also compatible with tools for measuring and manipulating neuronal activity at the single cell level, including two-photon calcium imaging and holographic optogenetics.

Motivated behaviors are essential for survival and can be disrupted in brain circuits, leading to various diseases such as addiction and obesity (*Kenny, 2011*; *Rossi and Stuber, 2018*; *Volkow et al., 2017*). Motivation in animal models is often assessed and quantified using multiple tasks that attempt to isolate distinct behavioral components such as appetitive and consummatory behaviors that can be the product of independent or overlapping brain circuits (*Panksepp, 1982*; *Robinson and Berridge, 1993*). To determine the role of brain circuits in distinct components of behavior, behavioral models with a high degree of experimental control and reproducibility are paramount as they can isolate components of behavior and limit variability across labs, subjects, and trials. There are a variety of approaches in freely moving rodents that model individual components of motivated behavior. Motivation is often modeled using operant responding on levers or nose pokes to earn a caloric reward or intracranial brain stimulation. A highly controlled version of operant responding includes retractable levers and retractable lick spouts to limit access of both operant and consummatory responses, respectively. In contrast, consummatory behaviors require measuring the volumetric amount of appetitive or aversive solutions. A particularly useful model is the brief-access task, which consists of trial-based presentations of one of multiple solutions, enabling recording of behavioral and neuronal responses to gradations of both rewarding and aversive solutions within a single session (*Boughter et al., 2002*; *Davis, 1973*; *Smith, 2001*). Despite the widespread use of these procedures in freely moving animals, there has been limited adaptation of these tasks for head-fixed rodents despite the advantages afforded by the of the head-fixed approach.

Here, we present OHRBETS (Open-Source Head-fixed Rodent Behavioral Experimental Training System), a low-cost, open-source platform of hardware and software for quantifying both operant and consummatory behavior in head-fixed mice. OHRBETS features the ability to precisely limit operant and consummatory behaviors during operant conditioning, replicating the retractable levers and spout aspects of the freely moving condition. OHRBETS has a multi-spout design that allows multiple solutions to be presented independently in a single behavioral session, enabling various behavioral experiments like probabilistic reinforcement tasks and choice behavior. The platform is also flexible and includes connectivity for additional customizable components. OHRBETS consists largely of 3D printed and low-cost components that reduce the total cost per system and maximizes reproducibility. Multiple research groups have developed models for head-fixed operant behavior with a variety of operant responses (*Bloem et al., 2022*; *Cui et al., 2017*; *Guo et al., 2014*; *Stephenson-Jones et al., 2020*; *Vollmer et al., 2021*; *Vollmer et al., 2022*), but many of these systems are built for a single experimental procedures with minimal publicly available resources needed for consistent replication. To assist with modification and reproduction of our system, we have created a GitHub repository (https://github.com/agordonfennell/OHRBETS; copy archived at *Gordon-Fennell, 2023*) that contains 3D models (also available through TinkerCad), assembly instructions, wiring diagrams, behavioral programs, and scripts for analysis. The OHRBETS platform will allow any investigator to harness the strength of head-fixed approaches to study the neurobiological underpinnings of motivation and related disease states while maintaining many crucial behavioral phenotypes established in freely moving animals.

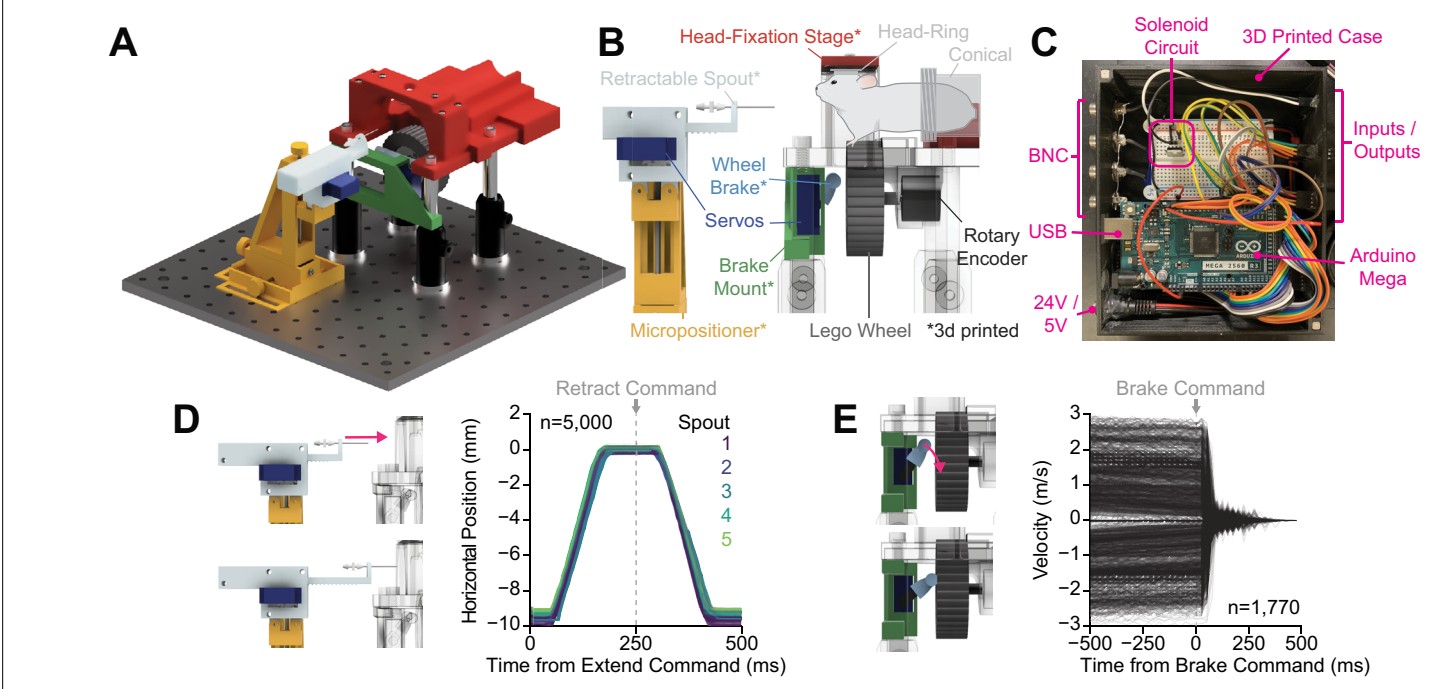

**Figure 1.** OHRBETS (Open-Source Head-fixed Rodent Behavioral Experimental Training System) operant conditioning. Overview of functionality for operant conditioning (additional, optional multi-solution functionality illustrated in *Figure 5*). (**A**) 3D rendering of OHRBETS. (**B**) Cartoon depicting the critical components of our system (* indicates 3D printed components). (**C**) Image of the Arduino-based microprocessor and custom enclosure used for controlling hardware and recording events. (**D**) Validation of our 3D printed retractable spout powered by a low-cost micro servo. Left: 3D rendering of the linear travel of the spout; right: horizontal position of the spout tip determined using DeepLabCut over time during 1000 extension/retractions with five unique retractable spout units. (**E**) Validation of our 3D printed wheel brake powered by a low-cost micro servo. 3D rendering of the rotational travel of the wheel brake (left); binned rotational velocity of the wheel produced by manual rotation before and after the brake is engaged (right).

The online version of this article includes the following figure supplement(s) for figure 1:

**Figure supplement 1.** Validation of retractable spout and wheel brake.

## Results

### OHRBETS overview

We developed OHRBETS, a low-cost, open-source system for head-fixed behaviors in mice (*Figure 1*, *Figure 1—figure supplement 1*). Our system consists of custom 3D printed and inexpensive, commercially available components bringing the total cost to around $600 for the operant-only version and around $1000 for the operant+multi-spout version. For head fixation, mice are implanted with a metal head ring and are easily and quickly secured on the head-fixed system for daily behavioral sessions (*Figure 1A–B*). To deliver solutions, including sucrose, we use gravity-fed tubing attached to a stainless steel lick spout that is gated by a solenoid (Parker). The position of the lick spout is controlled using a custom 3D printed micropositioner (*Backyard Brains, 2013*; *Hietanen et al., 2018*), and licks are detected using a capacitive touch sensing. To limit access to consumption, paralleling a retractable lick spout from the widely used freely moving operant assay, we used a linear actuator (adapted from *Buehler, 2016*) that is controlled using a 5 V micro servo for extending and retracting the spout (*Figure 1D*). A 43.2 mm diameter wheel (Lego, 86652c01) coupled to a rotary encoder is mounted underneath the mouse such that their forepaws' deflections left or right can serve as the operant response (*Aguillon-Rodriguez et al., 2021*). To limit access to operant responding, paralleling retractable levers used in the freely moving operant assay, we developed a wheel brake controlled via an additional micro servo (*Figure 1E*). All behavioral components are controlled by an Arduino Mega and the timing of events is relayed via serial communication and recorded using a Python program (*Figure 1C*). Our system is inexpensive and easily assembled following instructions freely available through our GitHub repository (https://github.com/agordonfennell/OHRBETS).

To characterize the effectiveness of our retractable spout and wheel brake, we conducted experiments to determine the timing and reliability of the hardware. We measured the linear travel of five sets of retractable spouts using high-speed video recording (200 fps) during 1000, 1 cm spout extensions/retractions and determined the position of the spout using DeepLabCut (*Mathis et al., 2018*; *Figure 2D*, *Figure 1—figure supplement 1A–B*). We found that the retractable spout follows a consistent and reliable pattern with >98% of extensions reaching a terminal position within 0.3 mm of each other in under 180 ms of the extension command (*Figure 1D*, *Figure 1—figure supplement 1A–E*). We measured the braking ability of four sets of wheel brakes by manually rotating the wheel at different rates in both directions and then programmatically engaging the brake (*Figure 1E*). The wheel brake rapidly stopped wheel rotation in 100% of trials, even with manual velocities that exceed that which a mouse can produce (*Figure 1E*). Furthermore, we analyzed the effectiveness of the brake to stop wheel rotation in data obtained during operant conditioning experiments and found that most mouse-generated rotations ceased in under 250 ms (*Figure 1—figure supplement 1F–I*). Together, these results indicate that OHRBETS produces reliable spout extension/retraction and wheel braking using inexpensive micro servos and 3D printed components, and therefore will effectively limit access to consummatory and operant responses during behavioral experiments.

## OHRBETS trained mice show multiple established characteristics of operant behavior observed in freely moving animals

We developed a training procedure that permits measuring operant conditioning in head-fixed mice, and we conducted a series of experiments to determine if operant behavior conducted with OHRBETS reproduces behavior seen in freely moving rodents (*Kliner et al., 1988*; *Reilly, 1999*; *Winger and Woods, 1985*). We trained head-fixed, water-restricted mice to perform operant conditioning in three stages: (1) free-access lick training, (2) retractable spout training, and (3) operant conditioning (Materials and methods). To measure the reproducibility of OHRBETS, all experiments were conducted using four independent operant-stage assemblies (referred to as box ID, data shown in supplements). To assess the potential differences between subsets of mice, we compared behaviors across each independent OHRBETS setup (box ID), sex, cohort (order of head-fixed and freely moving behavior). We compared the behaviors measured across box ID to determine if each setup was consistent enough to produce quantitatively similar behaviors despite the inherent variability associated with independent behavioral setups.

We trained mice on a single session of free-access lick training to facilitate licking from the spout and reduce stress associated with head fixation (*Figure 2—figure supplement 1*). Free-access lick training consisted of a 10 min session where each lick immediately triggered a delivery of ~1.5 μL of 10% sucrose which approximates free-access consumption from a standard lick spout (*Figure 2—figure supplement 1A*). During training, 100% of mice licked for sucrose throughout the session (*Figure 2—figure supplement 1B*). Similar to the standard freely moving free-access assay (*Johnson, 2018*; *Spector et al., 1998*), OHRBETS-trained mice licked in discrete licking bouts (*Figure 2—figure supplement 1C, D*). In the free-access lick training session, the total number of licks and the licking microstructure was consistent across sex, cohort, and box ID (*Figure 2—figure supplement 1E–P*), as well as across freely moving and head-fixed conditions (*Figure 2—figure supplement 1Q–V*).

Next, mice completed three sessions of retractable lick spout training - building the association between spout extension and the availability of reward to enhance the learning rate in subsequent operant conditioning (*Steinhauer et al., 1976*; *Figure 2A–E*, *Figure 2—figure supplement 2*). Each session consisted of 60 trials of spout extension, delivery of five pulses of 10% sucrose (~1.5 μL/ pulse, 200 ms inter-pulse interval), and a 5 s access period for liquid to be consumed during which an auditory tone (5 kHz, 80 dB) was presented. Mice licked to consume sucrose delivered on most trials with a short latency between spout extension and licking throughout each session (*Figure 2A–E*, *Figure 2—figure supplement 2*). By the third session of training, 31 out of 31 mice licked during 90% of trials (*Figure 2C*). Mice demonstrated a learned association between spout extension and a simultaneous auditory tone with the availability of sucrose, as they reduced their latency from spout extension to first lick across the three sessions of training (*Figure 2E*). No changes in the proportion of trials with a lick or the number of licks per trial over sessions were observed (*Figure 2C and D*). Female mice displayed a higher lick latency in response to spout extension compared to males on the first session of training, but the proportion of trials with a lick and the number of licks per trial was

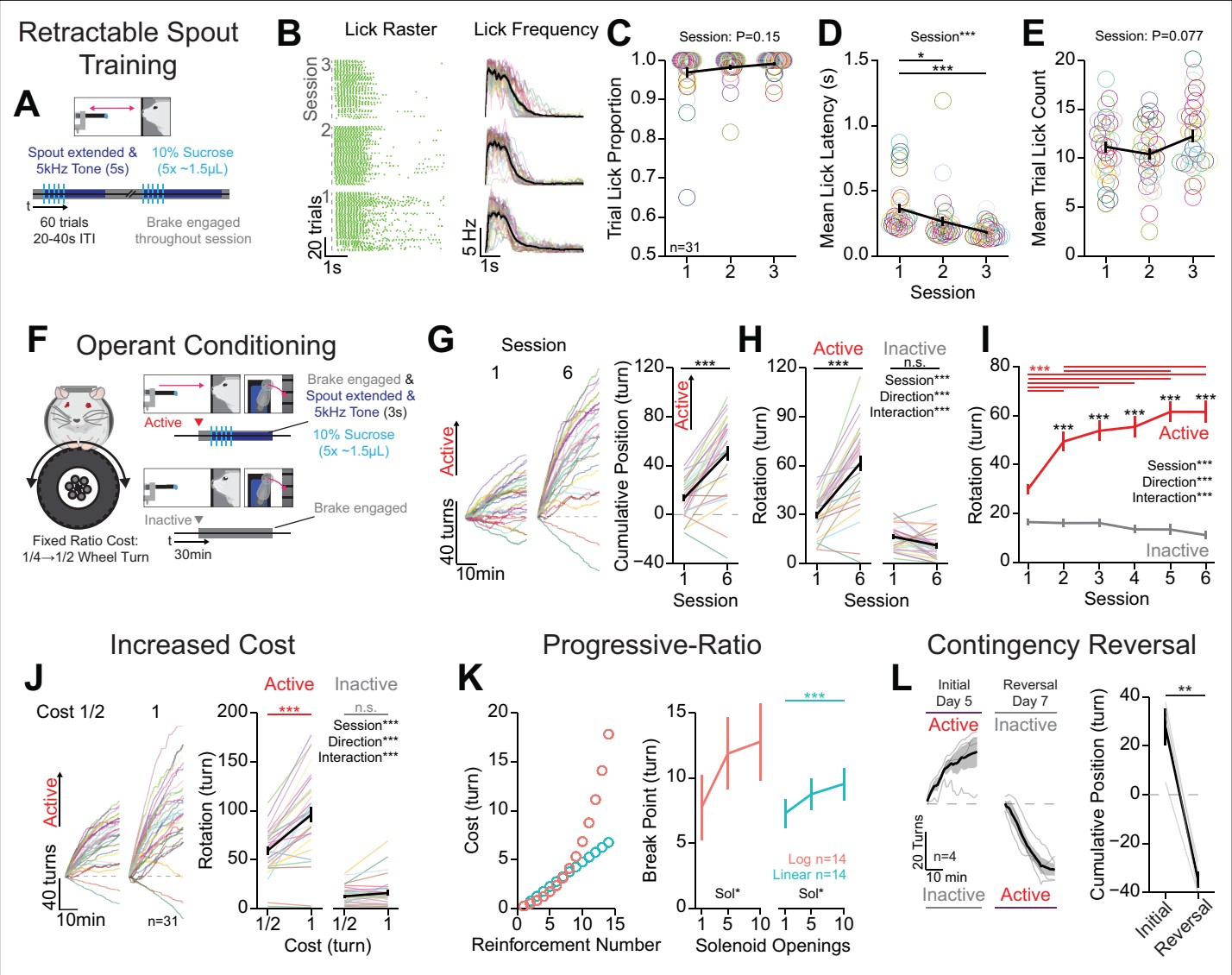

**Figure 2.** Mice rapidly learn head-fixed operant conditioning for sucrose and display operant behaviors established in freely moving experiments. (**A**) Cartoon depicting the task design for retractable spout training. (**B**) Licking behavior throughout retractable spout training; lick raster for a representative mouse with each lick represented as a tick (left); mean binned frequency of licks (right). (**C–E**) Summary of behavior throughout retractable spout training: proportion of trials with at least one lick (C; one-way repeated measures [RM] ANOVA, session effect, $F_{(2, 60)}=1.98$, p=0.15); mean latency from spout extension command to first lick on trials with a lick (D; one-way RM ANOVA, session effect, $F_{(2, 60)}=11.48$, ***p=6.01e-5; Tukey honest significant difference [HSD] post hoc, *p<0.05, ***p<0.001); mean number of licks within each 5 s access period (E; one-way RM ANOVA, session effect, $F_{(2, 60)}=2.68$, p=0.077). (**F**) Cartoon depicting the task design for operant conditioning. (**G**) Cumulative position of the wheel throughout the session (left) and at the conclusion of the session (right) on the first and sixth session of training (positive direction indicates rotation in the active direction; Welch two-sample t-test [paired, two-sided], $t_{(30)}=2.68$, ***p=5.8E-9). (**H, I**) Total rotation of the wheel throughout a session broken down based on direction on the first and sixth session of training (H; two-way RM ANOVA, session effect, $F_{(1, 30)}=70.15$, ***p=2.39e-9, rotation direction effect, $F_{(1,30)}=71.48$, ***p=1.96e-9, session × rotation direction interaction, $F_{(1,30)}=64.48$, ***p=5.8e-9, HSD post hoc, ***p<0.001) and across training sessions (I; two-way RM ANOVA, session effect, $F_{(5, 150)}=22.25$, ***p=1.22e-16, rotation direction effect, $F_{(1,30)}=78.16$, ***p=7.43e-10, session × rotation direction interaction, $F_{(5,150)}=23.54$, ***p=2.03e-17, HSD post hoc, ***p<0.001, red indicates comparisons across sessions for the active direction and black color indicates comparisons across rotation directions during the same session). (**J**) Cumulative position of the wheel throughout the last session (left), and the mean total rotation of the wheel in the last three sessions of fixed ratio 1/2 turn and 1 turn (two-way RM ANOVA, cost effect, $F_{(1,30)}=83.32$, ***p=3.66e-10, rotation direction effect, $F_{(1,30)}=79.69$, ***p=6e-10, cost × rotation direction interaction, $F_{(1,30)}=23.54$, ***p=2.03e-17, HSD post hoc, ***p<0.001). (**K**) Progressive-ratio schedule of reinforcement (left) and break points across different reward magnitudes set by the number of solenoid openings under a logarithmic schedule (one-way RM ANOVA, solenoid openings effect, $F_{(2,26)}=3.45$, *p=0.047), or linear schedule (one-way RM ANOVA, solenoid openings effect, $F_{(2,26)}=3.66$, *p=0.040, HSD post hoc, ***p<0.001). (**L**) Cumulative position of the wheel throughout the session (left) and at the conclusion of the session (right) on the last session of initial training and reversal training (Welch two-sample t-test [paired, two-sided], $t_{(3)}=9.64$,

*Figure 2 continued on next page*

*Figure 2 continued*

\*\*p=0.0024). (*Multi color lines and rings depict individual mice; black lines depict mean across mice; error bars depict standard error of the mean in all figure unless otherwise noted; see **Source data 1** for a complete presentation of the statistical results.*)

The online version of this article includes the following figure supplement(s) for figure 2:

**Figure supplement 1.** Quantification of behavior during head-fixed spout training.

**Figure supplement 2.** Quantification of behavior during head-fixed retractable spout training.

**Figure supplement 3.** Quantification of behavior during head-fixed operant conditioning for sucrose.

**Figure supplement 4.** Comparison of head-fixed and freely moving versions of operant conditioning.

---

not statistically different between males and females (*Figure 2—figure supplement 2E, H, K*). There were no differences in behavior across cohorts or behavioral systems (*Figure 2—figure supplement 2F, I, G, J, M*), aside from a significant interaction between cohort and session for the mean trial lick count (*Figure 2—figure supplement 2L*). Together, these data indicate that mice rapidly learn to lick for sucrose during discrete windows of access.

After free-access lick training and retractable spout training, water-restricted mice were operantly conditioned for sucrose (*Figure 2F–I*, *Video 1*, *Figure 2—figure supplement 3*). Operant conditioning consisted of six sessions of responding for 10% sucrose under fixed-ratio schedule (1/4 rotation for session 1; 1/2 rotation for sessions 2–6; *Figure 2F*; Materials and methods). To assess if mice learned the operant requirement, we examined whether mice increased responding in the active direction over sessions and exhibited a response bias for the active over the inactive response (*Heyser et al., 2000*). We found that mice learned to turn the wheel to obtain 10% sucrose in as little as one session, as 25/31 mice showed greater rotation in the active direction compared to the inactive direction (*Figure 2G*, session 1). By the sixth session of operant conditioning, 29/31 mice showed net rotation in the active direction (*Figure 2G*, data from all sessions shown in *Figure 2—figure supplement 3A*), that was the product of increased rotation in the active direction and no change in rotation in the inactive direction (*Figure 2H*). As a group, mice showed significantly more rotation in the active direction compared to the inactive direction starting at the second session (*Figure 2I*). Mice that were trained in each of the four boxes showed similar inter-lick intervals, trial lick counts, and latency to lick (*Figure 2—figure supplement 3B–D*). When analyzing behavioral data based on sex, cohort, and box ID, we found only minor differences in behavior (*Figure 2—figure supplement 3E–Y*). Notably, we found that over the course of training sessions, female mice exhibit a reduced total active rotation (*Figure 2—figure supplement 3H*), reduced total lick count (*Figure 2—figure supplement 3Q*), and reduced bias for rotation in the active direction (*Figure 2—figure supplement 3T*). These data indicate that mice rapidly exhibit operant responding for sucrose using OHRBETS, and this behavior is consistent across training history and behavioral setup with only minor differences observed between males and females.

Next, we determined if OHRBETS could reproduce other behaviors that have been established in freely moving rodents, including increased active responding following increased cost of reward (*Kliner et al., 1988*; *Winger and Woods, 1985*; *Figure 2J*), progressive ratio responding with a fixed reward magnitude (*Reilly, 1999*; *Sclafani and Ackroff, 2003*; *Winger and Woods, 1985*; *Figure 2K*), and reversal learning (*Forgays and Levin, 1959*; *Heyser et al., 2000*; *Klanker et al., 2015*; *Figure 2L*). To measure the relationship between cost and active response rate, after completing one session with a fixed ratio of 1/4 turn and five sessions of a fixed ratio of 1/2 turn (*Figure 2G–I*), we increased the fixed ratio to 1 turn and measured operant responding for four sessions. As observed in freely moving rodents (*Figure 2—figure supplement 4B*), when we increased the cost of reward, mice significantly increased responding in the active direction but not the inactive direction (*Figure 2J*, all sessions shown in *Figure 2—figure supplement 4A*), indicating

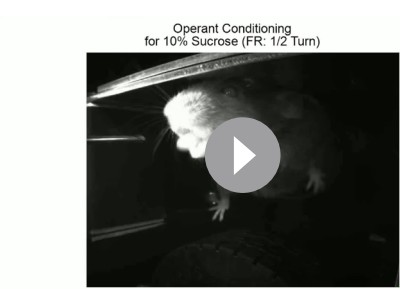

**Video 1.** Video of operant responding for 10% sucrose. Video of operant responding for 10% sucrose under a fixed ratio (FR) of 1/2 turn.

https://elifesciences.org/articles/86183/figures#video1

that they show flexible response rates as a function of reward cost (*Kliner et al., 1988*; *Winger and Woods, 1985*). Next, to measure the motivation to seek different reward magnitudes, we tested mice over multiple sessions of progressive ratio responding for sucrose of varying volumes (1, 5, 10 deliveries of ~1.5 µL of 10% sucrose, counterbalanced order). During progressive-ratio sessions, mice were tested with a linear or logarithmic reinforcement schedule, where the cost for each subsequent reinforcer was higher than the last (*Figure 2K*, left, Materials and methods). Under both schedules, mice responded for rewards during progressive ratio and displayed increased breakpoints for greater reward magnitude (*Reilly, 1999*; *Sclafani and Ackroff, 2003*; *Winger and Woods, 1985*; *Figure 2K*, right). To determine if mice can learn reversals in response contingency, we trained a naïve group of mice to perform operant responding with an initial rotational direction contingency for five sessions and then switched the contingency and allowed mice to re-learn over seven sessions. We found that mice displayed reversal learning, as they reversed the terminal cumulative position (initially active-initially inactive) following contingency reversal and training over seven sessions (*Heyser et al., 2000*; *Klanker et al., 2015*; *Figure 2L*). Finally, to directly compare behavior during head-fixed and freely moving versions of operant conditioning, we examined behavioral responding in the two tasks within the same mice (*Figure 2—figure supplement 4A, B*). We found that mice showed similar changes in response vigor with increased cost of reward (*Figure 2—figure supplement 4A–C*) and similar pattern of reduction in responding over the course of a session after the first 10 min (*Figure 2—figure supplement 4D*) but earned more liquid in the freely moving version of the task (*Figure 2—figure supplement 4E*). Together, these data indicate that mice display flexible operant behavior in our head-fixed system that is sensitive to the cost of reward, the magnitude of reward, and reward contingency, and produces behavior in a parallel manner to freely moving operant conditioning.

## OHRBETS-trained mice exhibit positive and negative reinforcement of operant responding during optogenetic stimulation of lateral hypothalamic GABAergic and glutamatergic neurons

After establishing that OHRBETS can measure operant responding for caloric rewards in mice, we determined if OHRBETS can measure operant responding to obtain or avoid optogenetic stimulation of brain circuits that have been previously established to be rewarding or aversive in freely moving rodents (*Chen et al., 2020*; *Jennings et al., 2015*; *Jennings et al., 2013*; *Rossi et al., 2019*). Optogenetic stimulation allows for temporally precise manipulations of genetically and spatially defined neuronal circuits enabling greater consistency of unconditioned stimuli delivery across a multitude of experimental conditions. We used optogenetic stimulation of lateral hypothalamic area (LHA) GABAergic neurons (LHA$^{GABA}$) as a appetitive unconditioned stimulus because activation of these neurons produces positive reinforcement (*Jennings et al., 2015*), and optogenetic stimulation of LHA glutamatergic neurons (LHA$^{Glut}$) as an aversive unconditioned stimulus because activation of these neurons is aversive (*Chen et al., 2020*; *Rossi et al., 2019*). To selectively manipulate LHA$^{GABA}$ and LHA$^{Glut}$ neurons, we expressed cre-dependent channelrhodopsin-2 (ChR2) or cre-dependent mCherry in the LHA of *Slc32a1*$^{Cre}$ (Vgat-Cre) or *Slc17a6*$^{Cre}$ (Vglut2-Cre) mice (*Vong et al., 2011*) and implanted bilateral optic fibers with a head ring to facilitate head fixation (*Figure 3A*, fiber placements depicted in *Figure 3—figure supplement 1A*, Materials and methods). Following incubation, mice were tested using freely moving and OHRBETS positive optogenetic reinforcement (responding to obtain stimulation) or negative optogenetic reinforcement (responding to remove stimulation) in counterbalanced order.

Mice displayed high levels of active responses to obtain optogenetic stimulation of LHA$^{GABA}$ neurons that was consistent across freely moving and head-fixed procedures (*Figure 3D–H*, *Video 2*, *Figure 3—figure supplement 2A–F*). We first trained mice to nose poke (fixed ratio 1 poke) or turn a wheel (fixed ratio 1/2 turn) to obtain optogenetic stimulation (1 s, 20 Hz, 5 ms pulse duration) of LHA$^{GABA}$ cells over four to five sessions (*Figure 3B and C*; training data shown in *Figure 3—figure supplement 2A–F*). Next, we measured operant responses for different stimulation frequencies by running mice through five sessions of positive optogenetic reinforcement with one of five stimulation frequencies (1, 5, 10, 20, 40 Hz) in counterbalanced order. On the last 20 Hz self-stimulation training session, Vgat-Cre mice expressing ChR2 in the LHA (LHA$^{GABA}$:ChR2) displayed high levels of operant responding for the active hole or active direction and displayed strong discrimination between active and inactive responses; Vgat-Cre mice expressing the mCherry control construct in the

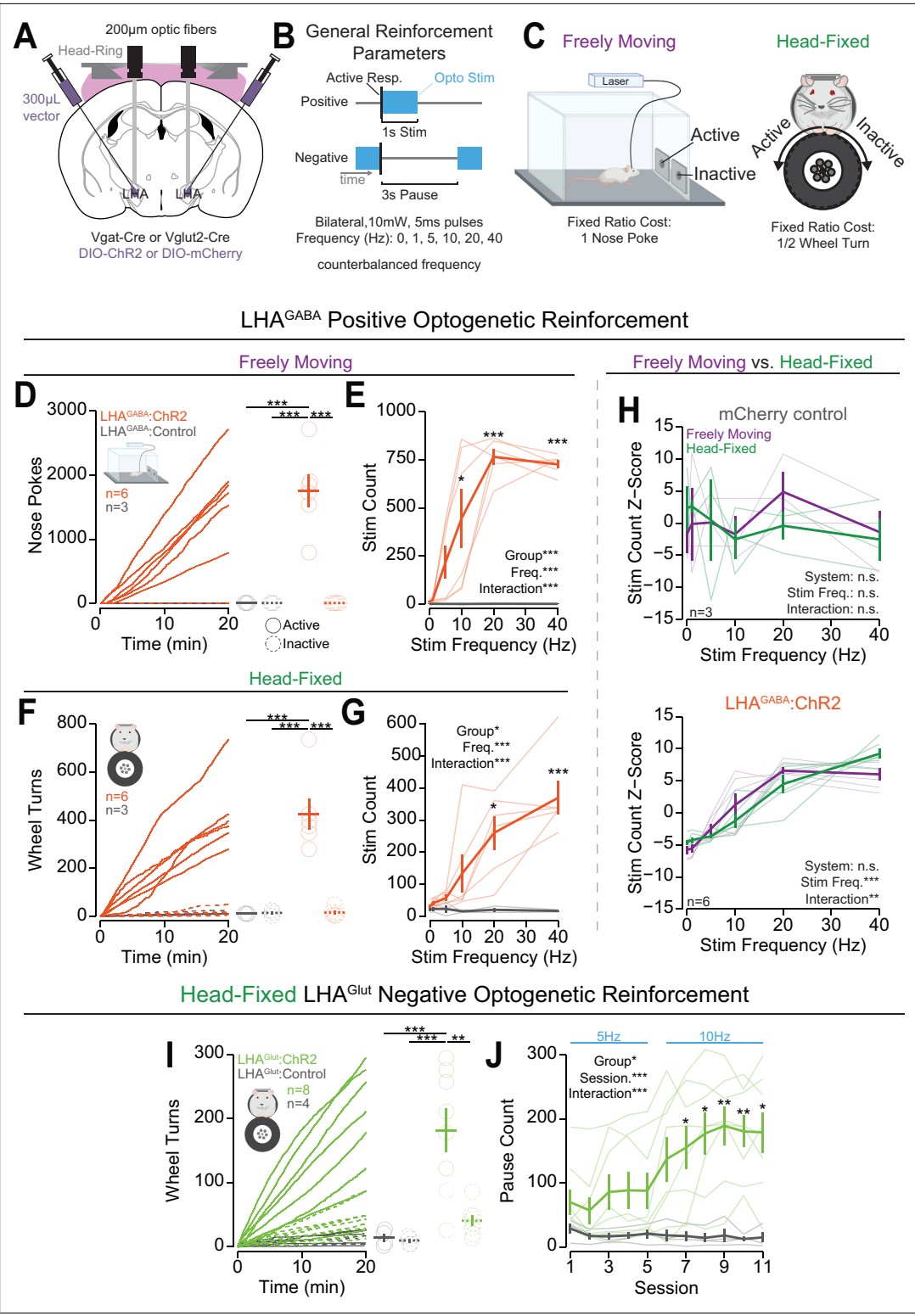

**Figure 3.** Head-fixed operant conditioning to obtain stimulation of lateral hypothalamic area GABAergic (LHA^GABA) neurons or avoid stimulation of LHA glutamatergic (LHA^Glut) neurons. (**A**) Approach, placements depicted in *Figure 3—figure supplement 1A*. (**B**) Diagram of the experimental approach for positive reinforcement conducted with LHA^GABA mice and negative reinforcement conducted with LHA^Glut mice. (**C**) Cartoon depicting the freely moving (left) and head-fixed (right) versions of the operant task. (**D**) Cumulative (left) and total (right) freely moving nose pokes under positive reinforcement for 40 Hz stimulation in LHA^GABA:ChR2 (red) and LHA^GABA:Control (gray) mice (two-way repeated measures [RM] ANOVA, group effect, $F_{(1,7)}=22.15$, **p=0.0022, response ID effect, $F_{(1,7)}=22.21$, **p=0.0022, group × response ID interaction, $F_{(1,7)}=22.19$, **p=0.0022, honest significant

*Figure 3 continued on next page*

*Figure 3 continued*

difference (HSD) post hoc, ***p<0.001). (**E**) Total number of stimulations earned under positive reinforcement within the freely moving task for different stimulation frequencies (two-way RM ANOVA, group effect, $F_{(1,7)}$=31.89, ***p=7.7e-4, frequency effect, $F_{(5,35)}$=12.25, ***p=6.7E-7, group × frequency interaction, $F_{(5,35)}$=12.33, ***p=6.8e-7, HSD post hoc, *p<0.05, ***p<0.001). (**F**) Cumulative (left) and total (right) head-fixed wheel turns under positive reinforcement for 40 Hz stimulation in LHA$^{GABA}$:ChR2 (red) and LHA$^{GABA}$:Control (gray) mice (two-way RM ANOVA, group effect, $F_{(1,7)}$=15.98, **p=0.0052, response ID effect, $F_{(1,7)}$=22.01, **p=0.0022, group × response ID interaction, $F_{(1,7)}$=22.78, **p=0.0020, HSD post hoc, ***p<0.001). (**G**) Total number of stimulations earned under positive reinforcement within the head-fixed task for different stimulation frequencies (two-way RM ANOVA, group effect, $F_{(1,7)}$=11.00, *p=0.013, frequency effect, $F_{(5,35)}$=9.00, ***p=1.44e-5, group × frequency interaction, $F_{(5,35)}$=9.50, ***p=8.6e-6, HSD post hoc, *p<0.05, ***p<0.001). (**H**) Comparisons of the z-score of the total number of stimulations across frequencies in freely moving (purple) and head-fixed (green) in LHA$^{GABA}$:Control mice (top, two-way RM ANOVA, system effect, $F_{(1,2)}$=0.00, p=1.00, frequency effect, $F_{(5,10)}$=0.27, p=0.92, system × frequency interaction, $F_{(5,10)}$=0.68, p=0.65, HSD post hoc) and LHA$^{GABA}$:ChR2 mice (bottom, two-way RM ANOVA, system effect, $F_{(1,5)}$=0.00, p=1.00, frequency effect, $F_{(5,25)}$=42.68, ***p=1.88e-11, system × frequency interaction, $F_{(5,25)}$=4.37, p=5.38e-3, HSD post hoc, no significant post hoc differences when comparing systems at the same stimulation frequency). (**I**) Cumulative rotation over a session under negative reinforcement for 5 Hz and 10 Hz stimulation in LHA$^{Glut}$:ChR2 (lime green) and LHA$^{Glut}$:Control (gray) mice (two-way RM ANOVA, group effect, $F_{(1,10)}$=13.78, **p=0.0040, response id effect, $F_{(1,10)}$=9.07, *p=0.013, group × response ID interaction, $F_{(1,10)}$=8.03, *p=0.018, HSD post hoc, **p<0.01, ***p<0.001). (**J**) Total pause count across all training sessions during negative reinforcement at the frequency indicated with blue text above the plot (two-way RM ANOVA, group effect, $F_{(1,10)}$=8.71, *p=0.015, session effect, $F_{(10,100)}$=5.67, ***p=1.18E-6, group × session interaction, $F_{(10,100)}$=6.88, ***p=4.22E-8, Bonferroni adjusted t-test post hoc, *p<0.05, **p<0.01). (*Faded lines and rings depict individual mice; asterisks above means indicate significant differences determined between stim count at a corresponding stim frequency or pause count at a corresponding session; asterisks above horizontal lines indicate significant difference determined between means indicated by edges of corresponding line; see **Source data 1** for a complete presentation of the statistical results.*)

The online version of this article includes the following figure supplement(s) for figure 3:

**Figure supplement 1.** Placement of optic fibers.

**Figure supplement 2.** Training data for operant conditioning to obtain or avoid optogenetic stimulation.

**Figure supplement 3.** Correlation between freely moving and head-fixed stimulation count during positive reinforcement for lateral hypothalamic area GABAergic (LHA$^{GABA}$) optogenetic stimulation.

**Figure supplement 4.** Optogenetic stimulation gates responding under positive and negative reinforcement.

---

LHA (LHA$^{GABA}$:Control) displayed little to no responding and did not discriminate between responses (*Figure 3D and F*). In both the freely moving and head-fixed conditions, LHA$^{GABA}$:ChR2 mice displayed greater active response rates for higher stimulation frequencies (*Figure 3E, G and H*) that were positively correlated across the two versions of the task (*Figure 3—figure supplement 3*). On the contrary, LHA$^{GABA}$:Control mice displayed no change in responding to changes in frequency and no correlation across the two procedures. Similar to freely moving positive optogenetic reinforcement (*Stuber et al., 2011*; *Witten et al., 2011*), mice rapidly ceased responding once optogenetic stimulation was withheld and resumed responding once optogenetic stimulation was reintroduced (*Figure 3—figure supplement 4*, left). These data indicate that OHRBETS can robustly elicit motivated behaviors to obtain rewarding optogenetic stimulation in a similar manner to freely moving rodent behavioral paradigms.

Mice displayed high levels of responding to avoid optogenetic stimulation of LHA$^{Glut}$ neurons under negative reinforcement during the head-fixed procedure but not the freely moving procedure. To elicit negative reinforcement (responses to cease an aversive stimulus) in the head-fixed procedure, we trained mice to turn a wheel to earn a 3 s pause of continuous stimulation of LHA$^{Glut}$ neurons at 5 Hz for sessions 1–5 and 10 Hz for sessions 6–11 (*Figure 3b*). Following training, Vglut2-Cre mice with expression of ChR2 in the LHA (LHA$^{Glut}$:ChR2) displayed high levels of responding in the active direction and strong discrimination between the active and inactive

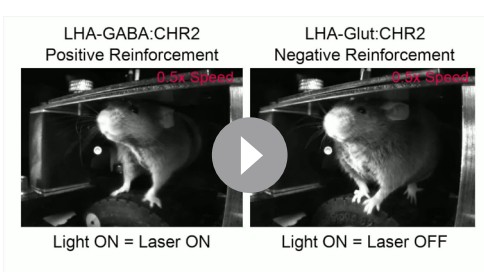

**Video 2.** Head-fixed operant conditioning to obtain stimulation of lateral hypothalamic area GABAergic (LHA$^{GABA}$) neurons or avoid stimulation of LHA glutamatergic (LHA$^{Glut}$) neurons. Videos showing responding for optogenetic stimulation of LHA$^{GABA}$ neurons under a positive reinforcement schedule (left) and responding for optogenetic stimulation of LHA$^{Glut}$ neurons under a negative reinforcement schedule (right). The LED near the center of the frame indicates when the optogenetic stimulation is turned on under positive reinforcement or when the optogenetic stimulation is turned off under negative reinforcement.
https://elifesciences.org/articles/86183/figures#video2

directions, while Vglut2-Cre mice with expression of mCherry control construct in the LHA (LHA$^{Glut}$:-Control) displayed little to no responding and no discrimination (*Figure 3I*). Over the course of training, LHA$^{Glut}$:ChR2 mice, but not LHA$^{Glut}$:Control mice, increased the number of pauses earned (*Figure 3J*). Compared to LHA$^{Glut}$:Control, LHA$^{Glut}$:ChR2 mice showed substantially higher active rotation as well as a moderately higher inactive rotation (*Figure 3—figure supplement 2J, L*). LHA$^{Glut}$:ChR2 mice acquired negative reinforcement behavior at a reduced rate compared to positive reinforcement for LHA$^{GABA}$ optogenetic stimulation (seven sessions to acquisition of negative reinforcement vs one session for positive reinforcement; *Figure 3J*, *Figure 3—figure supplement 2E, K*). Like positive reinforcement, LHA$^{Glut}$:ChR2 mice that were trained on negative reinforcement rapidly ceased responding when the optogenetic stimulation was removed and resumed responding when optogenetic stimulation was reintroduced (*Figure 3—figure supplement 4A*, right). To compare behavior in head-fixed to freely moving procedures, we trained the same mice under negative reinforcement in a freely moving procedure. We found that, compared to LHA$^{Glut}$:Control mice, LHA$^{Glut}$:ChR2 mice displayed suppressed amounts of active-responding, number of pauses earned, and inactive responding during the freely moving condition (*Figure 3—figure supplement 3G–I*). The discrepancy between acquisition of negative reinforcement in the head-fixed assay versus the freely moving assay could be attributed to the reduced range of actions mice can make in the head-fixed assay. These results indicate that OHRBETS can elicit responding under negative reinforcement using a simple stimulation procedure that is incapable of producing responding in traditional freely moving conditions.

## Head-fixed mice express real-time place preference and avoidance behaviors

We designed and tested a procedure analogous to measuring valance using real-time place testing (RTPT) (*Britt et al., 2012*; *Kravitz et al., 2012*; *Stamatakis and Stuber, 2012*; *Tye and Deisseroth, 2012*) in head-fixed mice which we name wheel time preference (WTP) (*Figure 4*). RTPT is extensively used to measure the appetitive or aversive characteristics of neuronal manipulations and importing this approach to head-fixed mice using WTP will permit new and exciting experiments not possible in freely moving subjects. With the same mice utilized for operant conditioning (Materials and methods), we used stimulation of LHA$^{GABA}$ neurons as a positive unconditioned stimulus and stimulation of LHA$^{Glut}$ neurons as a negative unconditioned stimulus because these two populations have been previously shown to drive real-time place preference and real-time place avoidance, respectively (*Jennings et al., 2015*; *Nieh et al., 2016*; *Rossi et al., 2019*; *Figure 4A*, for fiber placement see *Figure 3—figure supplement 1A*). Vgat-Cre and Vglut2-cre mice expressing mCherry were pooled and used as a single control group after observing no statistical differences in behavior between the two genotypes. In the standard RTPT, freely moving mice traverse a two-chamber arena in which they receive optogenetic stimulation when the mouse is located in one of the two chambers (*Figure 4B*, top). In WTP using OHRBETS, the response wheel was divided into two halves relative to the starting position of the wheel, one of which was paired with optogenetic stimulation (*Figure 4B*, bottom). To enhance the mouse's ability to determine their position on the wheel, we included two auditory tones of different frequencies (5 kHz and 10 kHz, 80 dB) that indicated the mouse's position in the two zones. The two-chamber RTPT and WTP assays offers a distinct advantage for comparing behavior across different versions of the assay because throughout the entire session duration the subject is in one of two states (stimulated or not), allowing for a one-to-one comparison of the amount of time stimulated over the fixed session duration. For both tasks, mice were initially habituated without stimulation for one session and then underwent RTPT/WTP over six sessions with frequency and chamber/wheel-zone pairing counterbalanced (*Figure 4C*). For the WTP, mice were initially trained without an auditory tone indicating the wheel zone. After initial training, we paired the wheel zones with auditory tones and found that mice exhibited more obvious preference/avoidance (*Figure 4—figure supplement 3A*), so in subsequent sessions these zone cues were added to the task design. Using this approach, we measured the similarity in preference/avoidance behavior with a range of rewarding and aversive stimulation magnitudes across freely moving RTPT and head-fixed WTP.

Mice expressed similar preference/avoidance behaviors during freely moving RTPT and head-fixed WTP procedures (*Figure 4D–E*, *Figure 4—figure supplement 1*). Specifically, mice expressing mCherry (LHA:Control mice) did not show preference nor aversion for the stimulation paired chamber/zone across all stimulation frequencies in both the freely moving and head-fixed procedures (*Figure 4D*

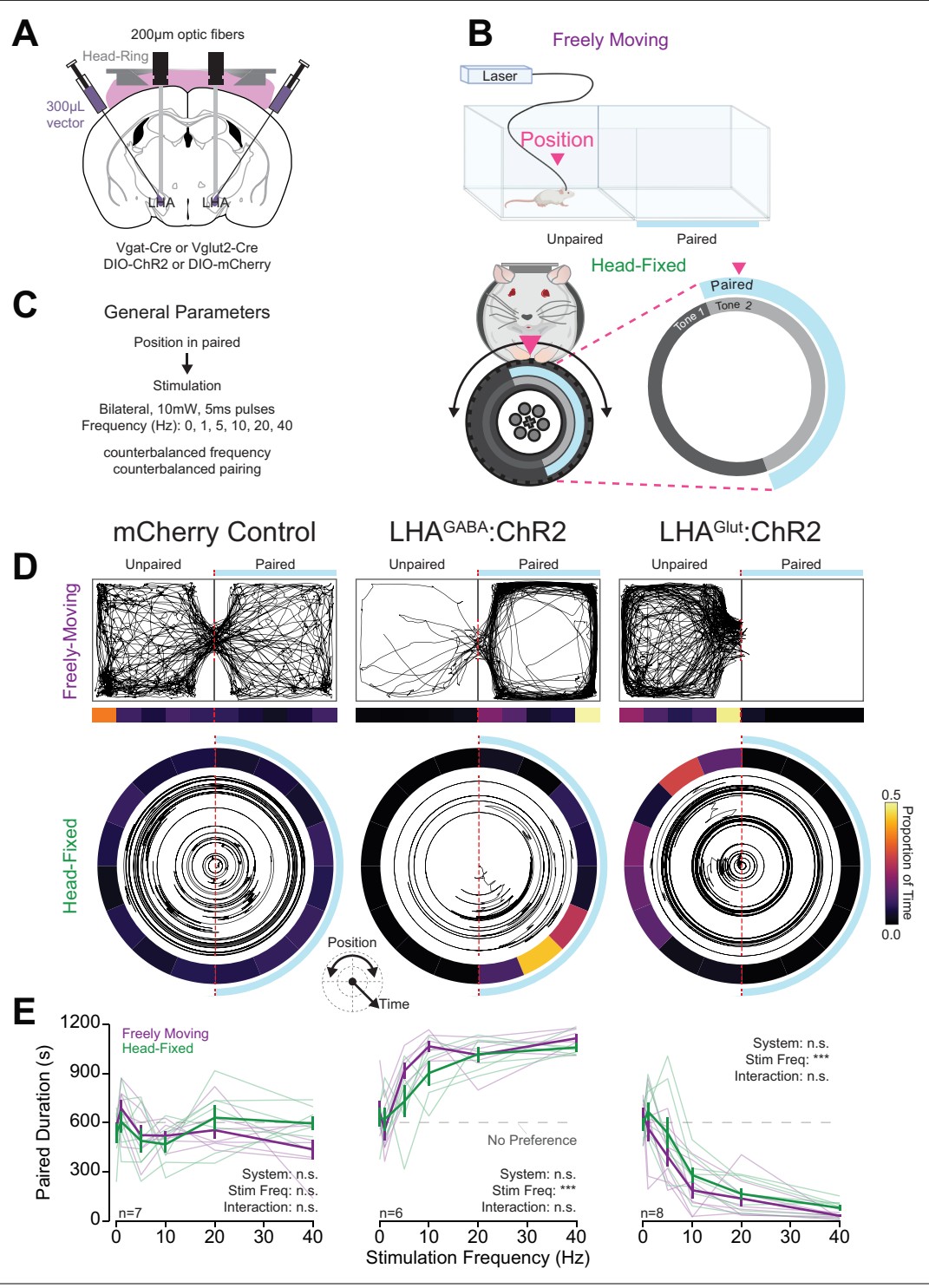

**Figure 4.** Head-fixed wheel time preference (WTP) and aversion associated with stimulation of lateral hypothalamic area (LHA) subpopulations mirrors freely moving behavior. (**A**) Approach, placements depicted in *Figure 3—figure supplement 1A*. (**B**) Cartoon depicting the freely moving and head-fixed versions of the operant task. In the head-fixed task, the mouse's position was determined relative to the position of the wheel and the mouse could rotate the wheel to navigate through the paired and unpaired zones. (**C**) Task design. (**D–F**) Behavior during the real-time place testing (RTPT) task; left column contains data from mCherry controls (both LHA^GABA:Control and LHA^Glut:Control), middle contains LHA^GABA:ChR2, right contains LHA^Glut:ChR2. (**D**) Representative traces of the mouse's position in the two-chamber arena in freely moving RTPT (top) and the

*Figure 4 continued on next page*

*Figure 4 continued*

position of the wheel over time in head-fixed RTPT (bottom). The right side of the arena or wheel was paired with optogenetic stimulation as indicated by the blue bar/arc. The proportion of time in binned areas of the arena or wheel are shown in the heat maps under or surrounding the traces (color scale represents the proportion of time in each position bin). (**E**) Amount of time spent in the paired zone during a 20 min (1200 s) session for varying frequencies; values above 600 s are indicative of preference, values below are indicative of avoidance. Colors represent the behavioral system as indicated in the left column (two-way repeated measures [RM] ANOVA, mCherry control: system effect, $F_{(1,6)}$=0.02, p=0.89, frequency effect, $F_{(5,30)}$=2.25, p=0.075, system × frequency interaction, $F_{(5,30)}$=1.42, p=0.25; LHA$^{GABA}$:ChR2: system effect, $F_{(1,5)}$=3.35, p=0.13, frequency effect, $F_{(5,25)}$=19.49, \*\*\*p=6.65e-8, system × frequency interaction, $F_{(5,25)}$=1.75, p=0.16; LHA$^{Glut}$:ChR2: system effect, $F_{(1,7)}$=4.01, p=0.085, frequency effect, $F_{(5,35)}$=66.75, \*\*\*p=6.73e-17, system × frequency interaction, $F_{(5,35)}$=1.18, p=0.34; *no honest significant difference (HSD) differences between systems were detected at corresponding stimulation frequencies; see Source data 1 for a complete presentation of the statistical results*).

The online version of this article includes the following figure supplement(s) for figure 4:

**Figure supplement 1.** Single subject data in the wheel time preference (WTP) and real-time place testing (RTPT) assays.

**Figure supplement 2.** Correlation of behavior measured with the wheel time preference (WTP) and real-time place testing (RTPT) assays.

**Figure supplement 3.** Behavior in the wheel time preference (WTP) with and without an auditory tone.

*and E*). On the contrary, LHA$^{GABA}$:ChR2 mice showed strong place preference while LHA$^{Glut}$:ChR2 mice showed strong place aversion for the paired chamber/zone with higher stimulation frequencies compared to lower stimulation frequencies (*Figure 4D and E*). There was no statistical difference between the amount of time in the paired chamber/zone in the freely moving and head-fixed versions of the task (*Figure 4D and E*). Furthermore, the time in the paired chamber/zone was correlated across freely moving RTPT and head-fixed WTP for LHA$^{GABA}$:ChR2 and LHA$^{Glut}$:ChR2 mice, but not LHA:Control mice (*Figure 4—figure supplement 2*). Together, these results indicate that the WTP task conducted with OHRBETS measures preference/avoidance behavior similar to freely moving procedures and provides a useful experimental approach for measuring the valence of stimuli.

## OHRBETS-trained mice display consummatory behaviors dependent on the concentration of appetitive and aversive solutions

Exposure to appetitive and aversive taste solutions provides an approach to measure neuronal correlates of appetitive and aversive events in addition to operant responding. Within-session consumption of unpredictable tastants allows for measuring a range of behavioral and neuronal responses to gradations in solution valence. We adapted OHRBETS to include a retractable, radial multi-spout consisting of five spouts (*Figure 5A*, *Video 3*). Using this system, we provided discrete access periods to one of five solutions with different concentrations in the same session with a task design adapted from the Davis Rig (*Davis, 1973*; *Smith, 2001*). Each behavioral session consisted of 100 trials with 3 s of free-access consumption separated by 5–10 s inter-trial intervals during which all spouts were in the retracted position (*Figure 5B*). Mice were given access to each of the solutions in pseudorandom order such that each solution was available two times every 10 trials. To control for modest spout effects (*Figure 5—figure supplement 1M-O*) and reduce prediction of the solution prior to tasting the solution (*Figure 5—figure supplement 4A-C*), we conducted the experiment counterbalanced over five sessions such that each spout was paired with each concentration (*Figure 5C, J and Q*). Using this approach, we measured within-session consumption of gradations in concentration of an appetitive solution (sucrose) and two aversive solutions (quinine and hypertonic sodium chloride [NaCl]).

Prior to behavioral training, mice were water-restricted to 80–90% baseline bodyweight (*Guo et al., 2014*). However, during behavioral sessions, multiple mice were able to consume enough fluid to maintain weight above 90% baseline body weight. Separate groups of mice were used for sucrose, quinine, and sodium chloride solution sets to control for training history. All groups of mice were initially conditioned on free-access licking in one to two sessions and then conditioned with the multi-spout procedure for three to seven sessions prior to five sessions of counterbalanced spout pairing (summarized in *Figure 5*). The licks measured using this approach approximate consumption, as total

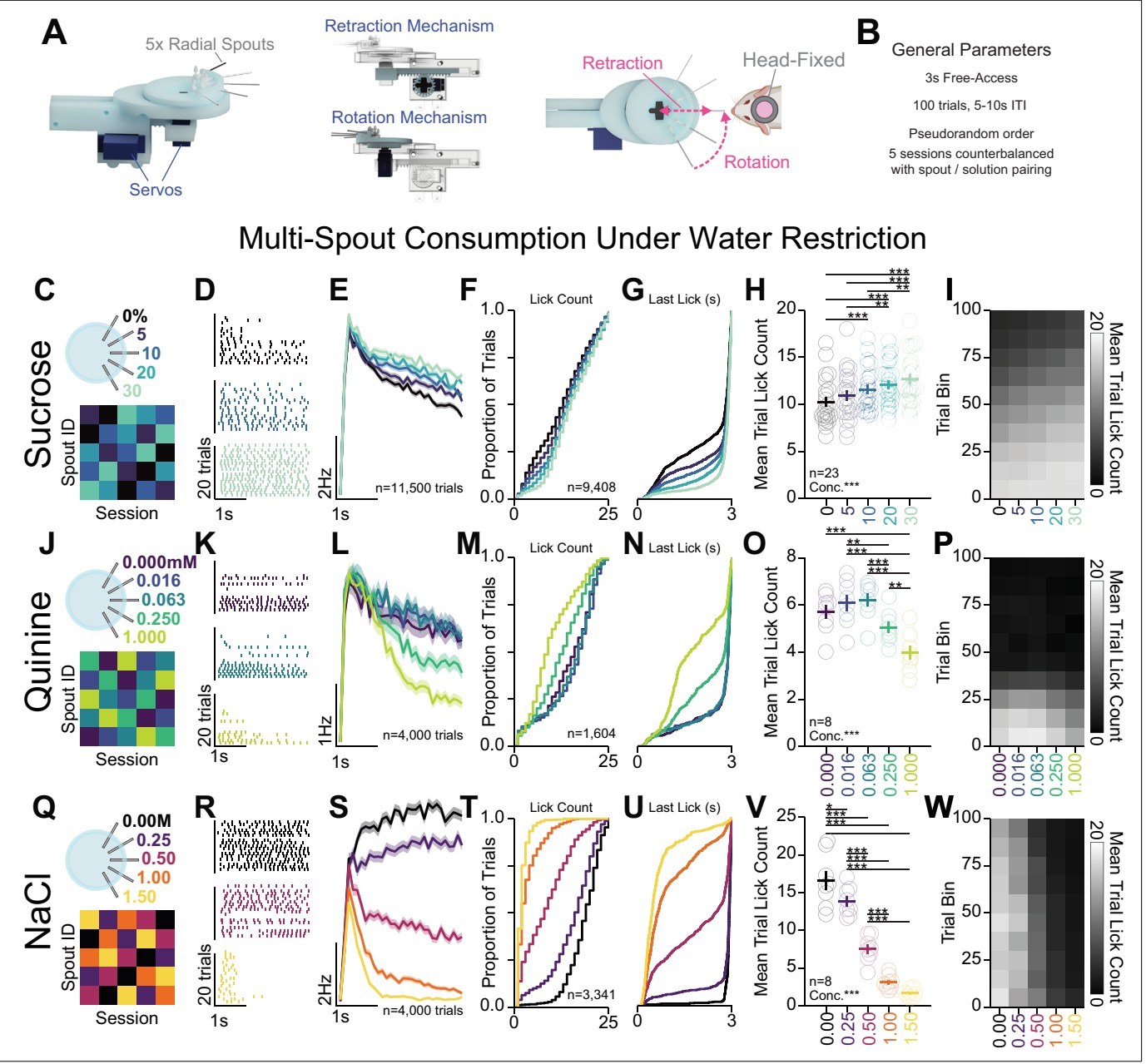

**Figure 5.** Head-fixed consumption of gradients of rewarding and aversive solutions during brief access. (**A**) 3D rendering of the multi-spout unit that retracts and rotates to allow brief-access periods to one of five lick spouts to the head-fixed mouse. (**B**) Task design. (**C–I**) Multi-spout consumption of a gradient of concentrations of sucrose data. (**C**) Procedure: mice received five sessions of 5× multi-spout counterbalanced to have each solution of each spout once. Colors represent concentrations of solution as defined in the label adjacent to the multi-spout cartoon. (**D**) Lick raster of a representative mouse depicting the licks for water, medium concentration, and high concentration during the 3 s access period. (**E**) Mean binned lick rate for all mice for each concentration. (**F–G**) Cumulative distribution of the number of licks in trials with a lick (**F**) and the time of the last lick within each licking bout (**G**). (**H**) The mean number of licks per trial for each concentration (one-way repeated measures [RM] ANOVA, concentration effect, $F_{(4,88)}$=19.18, ***p=2.26e-11, honest significant difference (HSD) post hoc, ***p<0.001, **p<0.01). (**I**) The mean number of licks for each concentration per trial binned by 10 trials over the course of the session. (**J–P**) same as (**C–I**), but for data from multi-spout consumption of a gradient of concentrations of quinine (one-way RM ANOVA, concentration effect, $F_{(4,28)}$=27.36, ***p=2.58E-9, HSD post hoc, ***p<0.001, **p<0.01). (**Q–W**) same as (**C–I**), but for data from multi-spout consumption of a gradient of concentrations of NaCl (one-way RM ANOVA, concentration effect, $F_{(4,28)}$=140.16, ***p=4.35e-18, HSD post hoc, ***p<0.001, **p<0.01, *p<0.05). (*Asterisks depict post hoc comparisons between concentrations indicated by edges of corresponding horizontal line; faded lines depict individual mice; see Source data 1 for a complete presentation of the statistical results.*)

The online version of this article includes the following figure supplement(s) for figure 5:

**Figure supplement 1.** Quantification of head-fixed consumption during brief access.

*Figure 5 continued on next page*

**Figure supplement 2.** Multi-spout consumption of different gradients of concentrations of quinine.

**Figure supplement 3.** Sex differences in head-fixed multi-spout consumption behavior.

**Figure supplement 4.** Potential anticipation of solution identity.

number of licks during each session is strongly correlated with weight in fluid consumed during the session (*Figure 5—figure supplement 1A*). Using this approach, we successfully measured a range of consumption responses with each set of solutions.

Mice displayed gradations in licking for different concentrations of sucrose, quinine, and sodium chloride (*Figure 5C–W*). For each solution set, licking bouts during the access period (representative session depicted in *Figure 5D, K, and R*, mean binned lick rate across all trials depicted in *Figure 5E, L, and S*) displayed inter-lick intervals similar to freely moving consumption (*Figure 5—figure supplement 1C*). Mice licking for gradations of sucrose (*Figure 5C–I*) showed a modest range of licking behavior where trials with higher concentrations of sucrose elicited a greater number of licks (*Figure 5F and H*) and longer time spent licking during the trial (*Figure 5G*). Mice licking for gradations of quinine (*Figure 5J–P*, *Figure 5—figure supplement 2*) showed a modest range of licking behavior where trials with higher concentrations of quinine elicited a lower number of licks (*Loney and Meyer, 2018*; *Figure 5M and O*) and shorter time spent licking during the trial (*Figure 5N*). Mice licking for gradations of NaCl (*Figure 5Q–W*, *Figure 5—figure supplement 3*) showed a large range of licking behavior where trials with higher concentrations of NaCl elicited a lower number of licks (*Figure 5T, V*) and shorter time spent licking during the trial (*Figure 5U*). Each solution set produced unique time courses of licking behavior over the course of the session (*Figure 5I, P, and W*, *Figure 5—figure supplement 1D*). Mice in the sucrose set started with high licking rates and showed a gradual satiation that resulted in decreased licking across all concentrations (*Figure 5I*, *Figure 5—figure supplement 1D*); mice in the quinine set started with high licking rates but rapidly dropped by around trial 40 across all concentrations (*Figure 5P*, *Figure 5—figure supplement 1D*); and mice in the NaCl set showed only a minor reduction in licking across all concentrations throughout the session (*Figure 5W*, *Figure 5—figure supplement 1D*). Comparing the total number of licks per session across the three sets of solutions revealed that mice displayed the highest number of licks during the sucrose set, then NaCl, then quinine (*Figure 5—figure supplement 1E*). Comparing task engagement using the proportion of trials with licking across the three sets of solutions, mice in the quinine set showed substantially lower proportion of trials with licks compared to mice in the sets for sucrose or NaCl (*Figure 5—figure supplement 1F*). Mice displayed little to no relationship between the number of licks in the session and weight of the mouse or amount of fluid consumed/provided on the previous session (*Figure 5—figure supplement 1G–J*). We also found that older mice displayed higher lick rates for sucrose (*Figure 5—figure supplement 1K, L*). Finally, we found no sex differences in task performance, except a lower proportion of trials with licking in female mice (*Figure 5—figure supplement 3*). Altogether, these data indicate that OHRBETS successfully measures a range of consumption behavior for differential concentrations of appetitive and aversive solutions.

Given that mice showed a smaller range of licking for gradations in quinine compared to NaCl, we further investigated licking behavior with additional sets of 1:4 serial dilutions of quinine with higher concentrations (starting concentration: low = 1 mM (*Figure 5J–P*), med = 5 mM, high = 10 mM) (*Figure 5—figure supplement 2*). Each quinine set produced a modest range of licking behavior with less licking for higher concentrations of quinine (*Figure 5—figure*

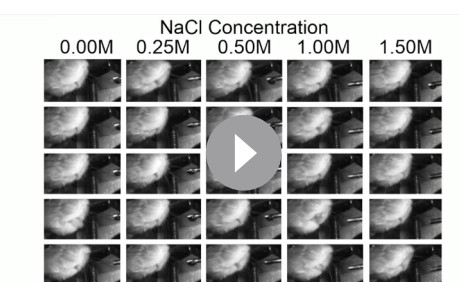

**Video 3.** Consumption behavior in the multi-NaCl assay under water restriction. Video shows licking behavior during the first 25 trials of the multi-spout assay for gradients of NaCl concentrations under water restriction. Each video depicts a single 3 s trial played back at half-speed. Videos are organized to display trials from top to bottom (earlier trials on the top), and NaCl concentration from left to right (lower concentrations on the left). However, concentrations were provided in pseudorandom order.
https://elifesciences.org/articles/86183/figures#video3

*supplement 2A–D*). Mice displayed a lower total licking in the high set compared to the med and low sets (*Figure 5—figure supplement 2E*), and mice in all sets showed similar task engagement as indicated by proportion of trials with licking (*Figure 5—figure supplement 2F*). Mice in all sets abruptly stopped licking part-way through the session (*Figure 5—figure supplement 2C*). Overall, each quinine set was capable of producing a range of licking behavior but failed to support licking throughout the entirety of the behavioral session.

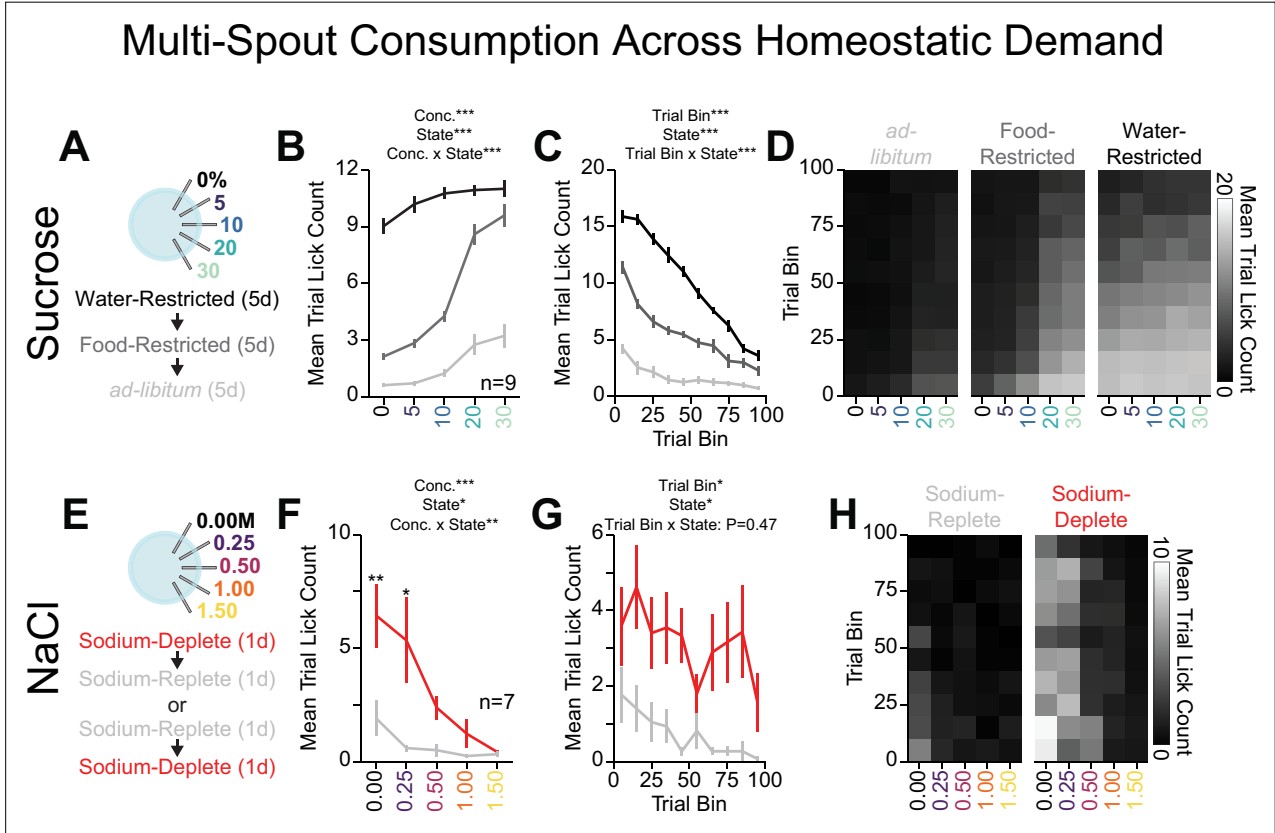

**Figure 6.** Homeostatic demand shifts within session consumption of gradients of sucrose and NaCl. (**A**) Procedure: Mice ran sequentially through water restriction, food restriction, and ad libitum states and during each state, mice received five sessions of multi-spout counterbalanced to have each concentration of sucrose on each spout once (two-way repeated measures [RM] ANOVA, concentration effect, $F_{(4,32)}$=157.23, ***p=1.48e-20, demand state, $F_{(2,16)}$=542.04, ***p=2e-15, concentration × demand state interaction, $F_{(8,64)}$=33.84, ***p=3.59e-20, honest significant difference [HSD] post hoc, every mean is significantly different from every other, except 30% sucrose consumption under food and water restriction). (**B**) The mean number of licks per trial for each concentration of sucrose in the ad libitum (light gray), food-restricted (dark gray), and water-restricted (black) states (two-way RM ANOVA, trial bin, $F_{(9,72)}$=149.63, ***p=6.19e-43, demand state, $F_{(2,16)}$=542.04, ***p=2e-15, trial bin × demand state interaction, $F_{(18,144)}$=35.43, ***p=7.32e-44). (**C**) Mean trial lick count across all concentrations of sucrose in bins of 10 trials across the session for each homeostatic state. (**D**) The mean number of licks for each concentration of sucrose per trial binned by 10 trials over the course of the session for each homeostatic state. (**E**) Procedure: In sodium-replete or sodium-deplete states in counterbalanced order, mice received one session of multi-spout with a gradient of concentrations of NaCl. The pairing of solution concentrations and spouts remained consistent. (**F**) The mean number of licks per trial for each concentration of NaCl in the sodium-replete (gray) and -deplete (red) states (two-way RM ANOVA, concentration effect, $F_{(4,24)}$=9.04, ***p=1.33e-4, demand state, $F_{(1,6)}$=13.19, *p=0.011, concentration × demand state interaction, $F_{(4,24)}$=4.41, **p=8.2e-3, HSD post hoc, **p<0.01, *p<0.05). (**G**) Mean trial lick count across all concentrations of NaCl in bins of 10 trials across the session for each homeostatic state (two-way RM ANOVA, trial bin, $F_{(9,54)}$=2.57, *p=0.016, demand state, $F_{(1,6)}$=12.76, *p=0.012 trial bin × demand state interaction, $F_{(9,54)}$=0.98, p=0.47). (**H**) The mean number of licks for each concentration of NaCl per trial binned by 10 trials over the course of the session for each homeostatic state. (*Asterisks above means indicate differences between homeostatic demand state at a corresponding concentration; see Source data 1 for a complete presentation of the statistical results.*)

The online version of this article includes the following figure supplement(s) for figure 6:

**Figure supplement 1.** Behavioral details for differences in consumption across homeostatic demand.

## Homeostatic demand shifts within-session consumption of gradients of sucrose and NaCl

To determine if OHRBETS multi-spout assay could detect shifts in consumption behavior following behavioral challenges, we measured consumption of a gradient of sucrose concentrations across homeostatic demand states. We trained mice in the multi-spout brief-access task for five sessions under water restriction, then five sessions under food restriction, and ending with five sessions under no restriction (ad libitum) (*Figure 6A*). We observed strong effects of restriction state on consumption behavior across sucrose concentrations (*Figure 6B–D*, *Figure 6—figure supplement 1A–E*). Most notably, mice showed a substantially larger range of licking behavior under food restriction compared to water restriction and ad-libitum (*Figure 6B*). Mice showed vastly different levels of total number of licks with the greatest number of licks for all concentrations under water restriction, then food restriction, then ad libitum (*Figure 6B and C*). Mice also displayed differences in licking rate throughout the session (*Figure 6C and D*). The minor scaling in licking across sucrose concentrations under water restriction compared to food restriction could indicate that the water component of the solutions is strongly appetitive under water restriction. Using OHRBETS, we measured changes in the relative consumption of concentrations of sucrose across homeostatic demand states that closely parallels the effect of homeostatic demand on sucrose consumption described in freely moving rodents (*Glendinning et al., 2002*; *Smith et al., 1992*; *Spector et al., 1998*).

To determine if our head-fixed multi-spout assay could detect shifts in consumption of NaCl, we measured consumption of a gradient of NaCl concentrations across sodium demand states. We first trained mice under water restriction (*Figure 5*) before allowing mice to return to ad libitum water. Next, we manipulated sodium appetite using furosemide injections followed by access to sodium-depleted chow (sodium-deplete) or standard chow (sodium-replete) and then measured consumption of a gradient of NaCl concentrations in our multi-spout assay over two sessions (counterbalanced order of sodium appetite state) (*Figure 6E*). Mice displayed greater licking under the sodium-deplete state compared to the sodium-replete state (*Figure 6F–H*, *Figure 6—figure supplement 1F–J*). Specifically, mice when sodium-deplete showed higher levels of licking for both water and 0.25 M NaCl. Mice displayed more licking throughout the session when sodium-deplete, indicating a heightened demand (*Figure 6G–H*). The increased licking for water when sodium-deplete can potentially be attributed to higher levels of thirst, as previously described (*Jalowiec, 1974*). Together, these results indicate that mice show a range of consummatory behaviors that are sensitive to homeostatic demand and that OHRBETS offers a platform for assessing shifts in consummatory drive in a reliable fashion in head-fixed mice.

## Light/dark cycle shifts within session consumption of gradients of sucrose

To characterize behavior across the circadian light/dark cycle, we measured consumption of a gradient of sucrose concentrations under food restriction during the dark cycle or light cycle in separate groups of mice (*Figure 7A*). During two sessions of free-access consumption, mice tested in the dark cycle consumed significantly more 10% sucrose compared to mice tested in the light cycle (*Figure 7B*; *Bainier et al., 2017*; *Smith, 2000*; *Tõnissaar et al., 2006*). Across eight sessions of the multi-spout assay, mice tested in the dark cycle licked more compared to mice tested in the light cycle (*Figure 7C*, left); however, over sessions 4–8 there was no effect of light cycle on licking (*Figure 7C*, right). Despite similar overall licking in the multi-spout assay, we found that experiments conducted during the light and dark cycle resulted in distinct licking across sucrose concentrations (*Figure 7E*; *Bainier et al., 2017*; *Tõnissaar et al., 2006*). Furthermore, compared to mice tested in the light cycle, mice tested during the dark cycle showed higher levels of consumption early in the session (*Figure 7F and G*). Together, these results indicate that the light/dark cycle affects sucrose consumption and testing mice in the light cycle leads to pronounced reductions in consumption in early training sessions.

## Comparing the reproducibility of the multi-spout brief-access task across independent laboratories

To determine if our system produces quantitatively similar consumption across labs, we compared behavior of food-restricted mice tested in the dark cycle trained on the multi-spout brief access to a gradient of sucrose concentrations obtained with our head-fixed system across independent labs

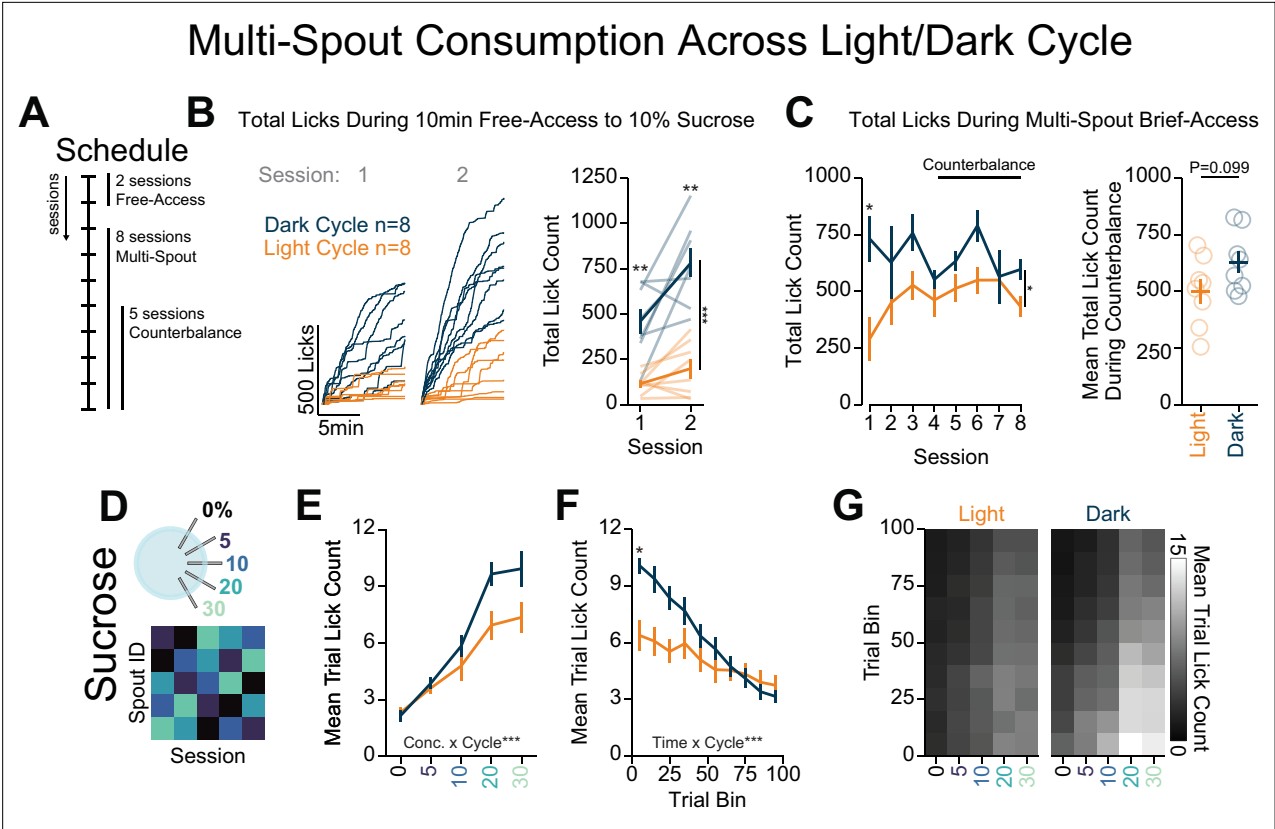

**Figure 7.** Light/dark cycle shifts within-session consumption of gradients of sucrose. (**A**) Schedule for behavioral sessions. (**B**) Licking behavior during two sessions of free-access licking for 10% sucrose displayed as cumulative licking (left) and total lick count during the session (right). During free-access training, mice tested in the dark cycle licked more than mice tested in the light cycle (two-way repeated measures [RM] ANOVA, cycle effect, $F_{(1,14)}$=58.52, ***p=2.30e-6, session effect, $F_{(1,14)}$=11.93, **p=3.87e-3, cycle × session interaction, $F_{(1,14)}$=4.13, p=6.15e-2, honest significant difference (HSD) post hoc, **p<0.01). (**C**) Total licking behavior during eight sessions of multi-spout brief access to a gradient of sucrose concentration (left) and mean over five counterbalance sessions (right). Mice tested in the dark cycle licked more than mice tested in the light cycle over all eight sessions (two-way RM ANOVA, cycle effect, $F_{(1,14)}$=5.24, *p=0.38, session effect, $F_{(7,98)}$=1.84, p=0.088, cycle × session interaction, $F_{(7,98)}$=1.97, p=0.066, HSD post hoc, *p<0.05), but not over the five counterbalanced sessions (sessions 4–8, Welch two-sample t-test [paired, two-sided], $t_{(14)}$=3.17, p=0.099). (**D**) Procedure: Mice were trained in five sessions of sucrose multi-spout counterbalanced to have each solution paired with each spout once. (**E**) Mean number of licks per trial for each concentration of sucrose for mice ran in the dark cycle (blue) and mice ran in the light cycle (orange) (two-way RM ANOVA, cycle effect, $F_{(1,14)}$=3.17, p=0.097, concentration effect, $F_{(4,56)}$=104, ***p=3.08e-25, cycle × concentration interaction, $F_{(4,56)}$=5.72, ***p=6.26e-4). (**F**) Mean trial lick count across all concentrations of sucrose in bins of 10 trials across the session (two-way RM ANOVA, cycle effect, $F_{(1,14)}$=3.15, p=0.097, time effect, $F_{(9,96)}$=42.6, ***p=4.36e-34, cycle × time interaction, $F_{(9,96)}$=9.19, ***p=1.3e-10, HSD post hoc *p<0.05). (**G**) The mean number of licks for each concentration of sucrose per trial binned by 10 trials over the course of the session. (*Asterisks above means indicate differences between mice tested in each cycle during the same session.*)

The online version of this article includes the following figure supplement(s) for figure 7:

**Figure supplement 1.** Comparison of multi-spout behavior across labs.

and geographic locations (*Figure 7—figure supplement 1*; data collected in the Stuber lab is shown in *Figure 6*, and data collected in the Roitman lab is shown in *Figure 7*). We observed qualitative differences in the binned licking rate over the 3 s access period (*Figure 7—figure supplement 1B*), with higher licking rate in mice tested in the Roitman lab near the onset of the access period. We also found that mice tested in the Roitman lab exhibited a small, but significant, reduction in inter-lick intervals compared to the Stuber lab (*Figure 7—figure supplement 1*). The source of these differences is not clear but could potentially be the product of differences in experimenter positioning of the spout resulting in subtle differences in licking patterns. However, despite these nominal differences, there were no statistical differences in the mean licking for each concentration of sucrose across labs (*Figure 7—figure supplement 1D*). These data indicate that our system produces similar consumption behavior when run in different labs, geographic locations, and experimenters.

# OHRBETS combined with fiber photometry to assess ventral striatal dopamine dynamics to multiple concentrations of rewarding and aversive solutions

To demonstrate the utility of the multi-spout assay run on OHRBETS, we performed simultaneous dual fiber photometry in the mesolimbic dopamine system during the multi-spout assay. The activity of ventral tegmental area dopamine neurons and the release of dopamine in the nucleus accumbens are well known to scale with relative reward value such that the most rewarding stimuli produces increases in dopamine release and the least rewarding stimuli produces modest decreases in dopamine release (*Eshel et al., 2015*; *Hajnal et al., 2004*; *Tobler et al., 2005*). We used multi-spout brief access to a gradient of an appetitive solution (sucrose) and an aversive solution (NaCl) to elicit a range of consummatory responses (*Figures 5 and 6*) while simultaneously recording dopamine dynamics in the medial nucleus accumbens shell (NAcShM) and lateral nucleus accumbens shell (NAcShL) (*Figure 8A*, placements shown in *Figure 8—figure supplement 6*). To record dopamine dynamics in the NAc, we expressed the dopamine sensor GRAB-DA (GRAB-DA1h [*Sun et al., 2018*] or GRAB-DA2m [*Sun et al., 2020*] in the NAcShM and NAcShL [counterbalanced hemispheres across mice]) of wild-type mice and implanted bilateral optic fibers and a head ring to facilitate head fixation (*Figure 8A*, Materials and methods). Mice were tested with multi-spout access to a gradient of sucrose concentrations under water restriction and food restriction, in counterbalanced order, and then a gradient of NaCl concentrations under water restriction (*Figure 8B*). Across each stage of the task, mice exhibited scaling in licking behavior that closely replicated data shown in *Figure 5* and *Figure 6*, *Figure 8—figure supplement 1*. During the multi-spout assay, we observed dynamics in dopamine signals in both the NAcShM and NAcShL during the consumption access period (*Figure 8C–O*). During consumption of sucrose under food restriction, where we observe a large range in licking across concentrations of sucrose (*Figure 8—figure supplement 1*, left), we measured strong scaling of GRAB-DA fluorescence in the NAcShL and moderate scaling in the NAcShM (representative mouse [*Figure 8D*]; mean fluorescence [*Figure 8E*]; mean fluorescence during access [*Figure 8F*]; cumulative distribution function [CDF] shown in *Figure 8—figure supplement 2A*). Specifically, we observed significantly higher responses in the NAcShL compared to the NAcShM at higher concentrations of sucrose (10%, 20%, and 30%). On a trial-by-trial basis, we observed a correlation between the amount of licking on a trial and GRAB-DA fluorescence (*Figure 8G*; CDF shown in *Figure 8—figure supplement 2B*). During consumption of sucrose under water restriction, where we observe high levels of licking but minimal range of licking across concentrations of sucrose (*Figure 8—figure supplement 1*, mid), we measured moderate scaling of GRAB-DA fluorescence in the NAcShL and little scaling in the NAcShM (representative mouse [*Figure 8H*]; mean fluorescence [*Figure 8I*]; mean fluorescence during access [*Figure 8J*]; CDF shown in *Figure 8—figure supplement 2C*). Specifically, mice displayed significantly higher GRAB-DA responses in the NAcShL compared to the NAcShM at higher concentrations of sucrose (*Figure 8J*). Like dynamics during food restriction, GRAB-DA fluorescence was positively correlated with licking within the trial (*Figure 8K*, CDF shown in *Figure 8—figure supplement 2D*). During consumption of the aversive tastant (NaCl) under water restriction, where we observed a large range of licking across concentrations of NaCl (*Figure 8—figure supplement 1*, right), we measured strong scaling of GRAB-DA fluorescence in both the NAcShL and NAcShM (representative mouse [*Figure 8L*], mean fluorescence [*Figure 8M*]; mean fluorescence during access [*Figure 8N*]; CDF shown in *Figure 8—figure supplement 2E*). Despite the interaction between solution and region of the NAc, there was a significantly higher GRAB-DA fluorescence in the NAcShL only during 0.25 M NaCl. Like other stages of the task, we observed a clear correlation between GRAB-DA fluorescence and licking during the trial (*Figure 8O*, CDF shown in *Figure 8—figure supplement 2F*). Taking advantage of the head-fixed preparation, we were able to record the activity of the NAcShL and NAcShM simultaneously and found a strong correlation in GRAB-DA fluorescence in the two regions across each stage of the task (*Figure 8—figure supplement 3*).

Interestingly, the NAcShM and NAcShL show a differential range of GRAB-DA fluorescence across each stage of the task. The NAcShM shows a disproportionately higher range of GRAB-DA fluorescence during multi-spout consumption of NaCl compared to the other stages of the task (*Figure 8—figure supplement 4B*). The higher range of dopamine release in the NAcShM during consumption of a range of aversive solutions compared to appetitive solutions could indicate a specialized role for the NAcShM in mediating behavioral responses to aversive stimuli as previously described for

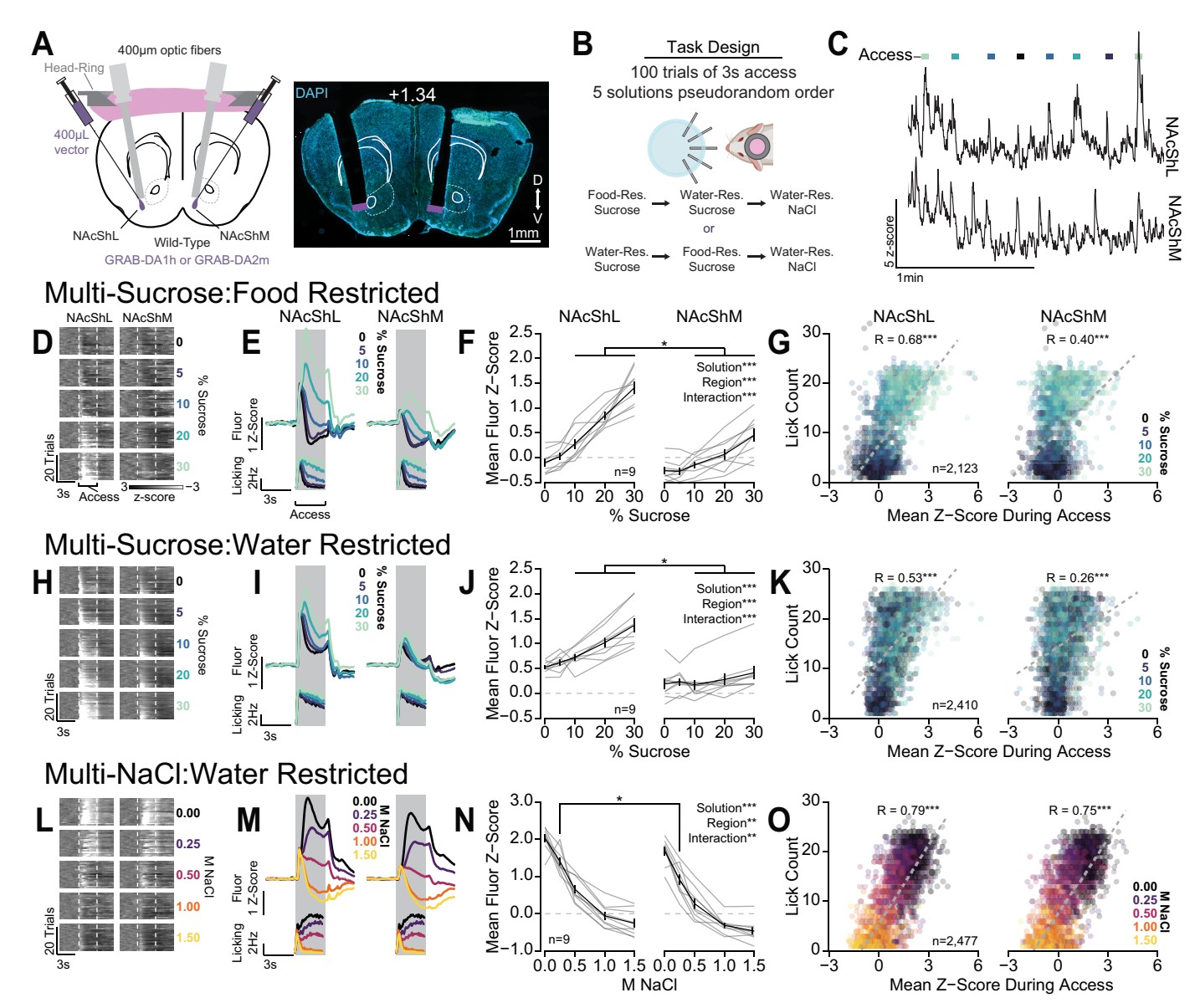

**Figure 8.** Differential dopamine dynamics during multi-spout consumption behavior. (**A**) Approach for simultaneously recording dopamine dynamics in the lateral nucleus accumbens shell (NAcShL) and medial nucleus accumbens shell (NAcShM) (left), and representative placements of optic fibers overlaying the NAcSh (white numerical value indicates AP position relative to bregma). (**B**) Task design and schedule of experiment. (**C**) Representative trace of simultaneous GRAB-DA fluorescence in the NAcShM and NAcShL during multi-spout access to sucrose under food restriction (*lines on top indicate access periods, color indicates sucrose concentration*). (**D–G**) Dopamine dynamics during multi-sucrose under food restriction: (**D**) Representative heat map of GRAB-DA fluorescence over time during each trial sorted by sucrose concentration (trials averaged over three sessions of recording, earliest trails depicted on bottom). (**E**) Perievent time histograms of mean GRAB-DA fluorescence (top) and licks (bottom) separated by sucrose concentration. (**F**) Mean fluorescence *z*-score during access period indicating strong scaling in the NAcShL (left) and weak scaling in the NAcShM (right) (two-way repeated measures [RM] ANOVA, concentration effect, $F_{(4,32)}$=71.29, ***p=1.77e-15, brain region effect, $F_{(1,8)}$=51.63, ***p=9.38e-5, concentration × brain region interaction, $F_{(4,32)}$=14.94, ***p=5.46e-7, honest significant difference (HSD) post hoc, *p<0.05 NAcShL vs NAcShM at same concentration). (**G**) On individual trials, the mean *z*-score during access correlates with licking in both the NAcShL (left) and NAcShM (right) (color depicts the solution concentration) (Pearson's product-moment correlation, NAcShL, *r*=0.68, ***p=2.06e-292, NacShM, *r*=0.40, ***p=5.08e-82). (**H–K**) Same as D–G for dopamine dynamics during multi-sucrose under water restriction. (**H**) Representative heat map of GRAB-DA fluorescence over time during each trial sorted by sucrose concentration (trials averaged over three sessions of recording, earliest trails depicted on bottom). (**I**) Perievent time histograms of mean GRAB-DA fluorescence (top) and licks (bottom) separated by sucrose concentration. (**J**) Mean fluorescence *z*-score during access period indicating moderate scaling in the NAcShL (left) and weak scaling in the NAcShM (right) (two-way RM ANOVA, concentration effect, $F_{(4,32)}$=20.81, ***p=1.57e-8, brain region effect, $F_{(1,8)}$=27.82, ***p=7.51e-4, concentration × brain region interaction,

*Figure 8 continued on next page*

*Figure 8 continued*

$F_{(4,32)}$=11.61, ***p=6.16e-6, HSD post hoc, *p<0.05 NAcShL vs NAcShM at same concentration). (**K**) On individual trials, the mean *z*-score during access correlates with licking in both the NAcShL (left) and NAcShM (right) (color depicts the solution concentration) (Pearson's product-moment correlation, NAcShL, *r*=0.53, ***p=4.53e-173, NacShM, *r*=0.26, ***p=1.45e-37). (**L–O**) Same as D–G for dopamine dynamics during multi-NaCl under water restriction. (**L**) Representative heat map of GRAB-DA fluorescence over time during each trial sorted by NaCl concentration (trials averaged over three sessions of recording, earliest trials depicted on bottom). (**M**) Perievent time histograms of mean GRAB-DA fluorescence (top) and licks (bottom) separated by NaCl concentration. (**N**) Mean fluorescence *z*-score during access period indicating strong scaling in the NAcShL (left) and strong scaling in the NAcShM (right) (two-way RM ANOVA, concentration effect, $F_{(4,32)}$=123.48, ***p=5.66e-19, brain region effect, $F_{(1,8)}$=11.28, **p=0.010, concentration × brain region interaction, $F_{(4,32)}$=4.53, **p=0.0052, HSD post hoc, *p<0.05 NAcShL vs NAcShM at same concentration). (**O**) On individual trials, the mean *z*-score during access correlates with licking in both the NAcShL (left) and NAcShM (right) (color depicts the solution concentration) (Pearson's product-moment correlation, NAcShL, *r*=0.79, ***p<2.23e-308, NacShM, *r*=0.75, ***p<2.23e-308).

The online version of this article includes the following figure supplement(s) for figure 8:

**Figure supplement 1.** Multi-spout licking behavior.

**Figure supplement 2.** Cumulative distribution functions of GRAB-DA responses in the NAcSh during multi-spout consumption behavior.

**Figure supplement 3.** Linear correlation of dopamine dynamics during multi-spout consumption.

**Figure supplement 4.** Range of licking and NAcSh dopamine signals during multi-spout consumption behavior.

**Figure supplement 5.** Representative full-session traces.

**Figure supplement 6.** Fiber placements for fiber photometry.

footshock conditioning (*de Jong et al., 2019*). OHRBETS allowed us to isolate consumption behavior in response to a range of rewarding and aversive solutions while performing dual site fiber photometry and revealed robust dopamine responses that scales with solution value and consumption. Furthermore, these data indicate that OHRBETS is highly compatible with neural recording and manipulation techniques that would be challenging with freely moving behavioral designs.

## Discussion

OHRBETS is a customizable, inexpensive system for head-fixed behavior in mice that enables a variety of behavioral experiments, including operant conditioning, RTPT, and multi-solution brief-access consumption, accurately replicating behaviors in freely moving. These data demonstrate that a diverse set of operant and consummatory behaviors are compatible with head-fixed procedures run with a single hardware setup and will serve as a resource for future investigations into these behaviors using neuroscience approaches that rely on head fixation.

Behavior measures within our head-fixed adaptations of freely moving operant assays reproduce many important phenotypes originally characterized in freely moving behavior. Mice rapidly learn operant responding for sucrose and then flexibly express responding as a function of reward cost and reward size (*Kliner et al., 1988*; *Reilly, 1999*; *Sclafani and Ackroff, 2003*; *Winger and Woods, 1985*). Using optogenetic stimulation, which offers tighter control over the precise magnitude and timing of appetitive and aversive states, mice exhibited quantitatively similar positive optogenetic reinforcement, preference, and avoidance behavior with our head-fixed and freely moving approaches. The ability to conduct operant reinforcement and WTP with a single setup is particularly useful in measures of valence-related neural circuits, but these results also imply that the head-fixed WTP procedure could be used to test the appetitive and aversive quality of other stimuli that are challenging to test in freely moving conditions including discrete somatosensory stimuli. Taken together, our results establish that our behavioral system produces robust, reproducible operant behavior consistent with the commonly employed freely moving counterparts.

Despite the quantitatively similar preference and aversion we measured between the head-fixed WTP assay and freely moving RTPT assay, there exists multiple differences between these assays that could influence behavioral results and their interpretation. Similar to the RTPT assay, the WTP assay assesses valence through the use of two mutually exclusive states (e.g. optogenetic stimulation vs non-stimulation). However, the WTP assay does not replicate the spatial components of the RTPT assay (*Gordon-Fennell and Stuber, 2021*). The head-fixed and freely moving assays almost certainly rely on different neuronal circuits for completing the task, as the head-fixed WTP does not contain the spatial contextual cues that are inherent to freely moving RTPT and instead relies on discrete auditory

cues in addition to the internal state of the subject. Due in part to this distinct difference, we do not expect that all circuit manipulations will produce comparable behavior across the head-fixed WTP and freely moving RTPT even though optogenetic stimulation of LHA$^{GABA}$ and LHA$^{Glut}$ produced similar preference and avoidance behaviors within these two assays. Therefore, while WTP conducted using OHRBETS offers the ability for new and exciting experiments to assess valence of stimuli in head-fixed mice, the relation of these results to the results of RTPT should be interpreted with caution.

In addition to the operant conditioning experiments, our system can facilitate multi-solution brief-access experiments for studying consummatory behavior. In our task, mice show consumption of a gradient of sucrose, quinine, and NaCl concentrations that closely matches behavior with the freely moving version of the task (*Corbit and Luschei, 1969*; *Coss et al., 2022*; *Garcia et al., 2020*; *Glendinning et al., 2002*; *St John et al., 1994*; *Loney and Meyer, 2018*; *Smith et al., 1992*; *Villavicencio et al., 2018*). Licking increased monotonically with increased concentrations of sucrose across all homeostatic states (*Garcia et al., 2020*; *Glendinning et al., 2002*; *Smith et al., 1992*; *Spector et al., 1998*). However, homeostatic demand states produced pronounced differences in the range of consumption behavior across sucrose concentration, as food restriction produced a substantially larger range of licking behavior compared to water restriction. One unexpected finding was that mice showed vastly different behavior when licking for the aversive tastants quinine and hypertonic NaCl. When licking for quinine, mice abruptly ceased consumption for all concentrations mid-way through the session. On the other hand, when licking for NaCl, mice continue to consume large amounts of low concentrations of NaCl throughout the entire session. These results may be explained by an additive effect of quinine that builds in aversion over trials and results in a lingering bitter taste (*Leach and Noble, 1986*) that attenuates motivation to initiate consumption. During the NaCl sessions, NaCl may stimulate thirst (*Kraly et al., 1995*; *O'KELLY, 1954*; *Stricker et al., 2002*) resulting in enhanced motivation to consume water. Thus, the multi-spout brief-access task with gradients of NaCl can be a uniquely advantageous approach for eliciting a high number of strongly aversive events in response to the highest concentrations of NaCl (1.0 and 1.5 M) while continuing to sustain behavioral engagement. Changes in task design could improve performance during the quinine task, such as including water rinse trials between each quinine trial (*Loney and Meyer, 2018*). In addition to using a gradient of solution concentrations, any number of combinations of tastants could be used to study a whole host of behavioral phenomena including innate and conditioned consumption behaviors.

One potential confound of the radial head design for the multi-spout brief-access experiments is that subjects may be able to learn the relationship between the rotational position of the radial head and the solution in order to use this information to predict solutions before tasting them. We attempted to mitigate this by counterbalancing the spout ID and solution pairings over sessions and conducting experiments in the dark, but we still observed modest reductions in the proportion of trials with licking at the highest concentrations of NaCl (*Figure 5—figure supplement 4A*) which may indicate that mice are able to learn to predict the solution identity over the course of a behavioral session. This effect is not universal, as we did not observe differences in the proportion of trials with licking during sessions with gradients of sucrose and quinine concentrations (*Figure 5—figure supplement 4B–C*). To reduce the chance that anticipatory information is shaping the neuronal correlates associated with different solution trials and ensure that neuronal signals are in response to tasting the solution, we recommend filtering trials to include only trials with licking as we performed in *Figure 8*. Minor adjustments to the approach including increased distance of retraction/extension with enhanced light blockage could further reduce the ability of the subject to predict the solution. Future designs could also take advantage of needle bundles to completely remove the solution prediction (*Perez et al., 2013*). Overall the data indicates that under some circumstances, mice are able to predict the solution which should be carefully considered when analyzing behavior and neuronal activity.

Using our multi-spout brief-access task in conjunction with GRAB-DA fiber photometry, we observed dopamine dynamics that positively correlated with relative solution value and consumption. Previous studies have revealed that dopamine release in the ventral striatum (*Hajnal et al., 2004*) and dopamine neuron activity scales with reward magnitude (*Eshel et al., 2015*; *Tobler et al., 2005*). We found that dopamine release in these subregions scales with the relative value of the solution being consumed and the amount of concurrent consumption and is strongly influenced by solutions present in a session and the mouse's homeostatic demand state. We observed differential scaling in the NAcShM and NAcShL across solution sets and homeostatic demand. Dopamine release in the NAcShL

showed higher amplitude increases to the most rewarding solutions, especially during consumption of sucrose in both water- and food-deprived states (*Figure 8*). While we observed greater GRAB-DA signals in the NAcShL compared to the NAcShM, it is important to consider that the GRAB-DA signal reflects changes in dopamine release rather than absolute levels of dopamine. This means that the greater signals we observed in the NAcShL could be the consequence of greater dopamine release or a lower dopamine tone at baseline. The range of dopamine release in the NAcShL closely matches the range of consumption behavior within each stage of the task, while dopamine release in the NAcShM appears disproportionately higher during consumption of gradients of NaCl. If we assume that the range of values is greater during multi-spout consumption of gradients of NaCl compared to gradients of sucrose, as indicated by a greater range in licking behavior (*Figure 8—figure supplement 4*), then the greater range of dopamine release in the NAcShM could imply that dopamine release in this structure tracks value. Alternatively, this result could indicate a specific role of dopamine release in the NAcShM that corresponds to shaping behavior or learning in the face of aversive events (*de Jong et al., 2019*). By conducting these experiments using OHRBETS, we removed approach behaviors that occur prior to consumption and isolated neuronal responses specifically during consumption (*Chen et al., 2022*) without interference of activity ramps observed in freely moving behavioral designs (*Howe et al., 2013*). Future experiments are necessary to reveal the specific contribution of licking, taste, and value to widespread dopaminergic signals and how these signals causally influence ongoing consumption or learning.

Eliminating locomotion improves compatibility with many standard neuroscience approaches including optogenetics, fiber photometry, electrophysiology, and calcium imaging. To prevent twisting of tethers, each of these approaches requires a commutator in freely moving conditions, but with head fixation the need for a commutator is eliminated. This facilitates multiplexed experiments with simultaneous use of multiple approaches that each rely on independent tethers without the risk of weighing down the animal, tangling, or twisting to the point of affecting task performance. For fiber photometry, fixing the animal dramatically reduces motion artifacts, thereby reducing the need for an isosbestic to correct for motion (*Figure 8—figure supplement 5*). This opens the ability to conduct experiments with fluorescence biosensors without known isosbestic points or without true isosbestic points. Recent advances in optical imaging have opened up new approaches in freely moving animals, but the cutting edge of optical technologies will typically start with tabletop microscopes. Using head-fixed models permits users to embrace cutting edge imaging technologies without waiting for further advances to bring the technology into freely moving animals. The use of OHRBETS allows for enhanced compatibility with a variety of neuroscience technologies and will enable novel, multiplexed experiments that would be difficult or impossible to conduct in freely moving animals.

While head-fixed experiments offer many advantages, they come with important caveats, limitations, and experimental design considerations. Head fixation can be acutely stressful to mice and causes increased levels of circulating stress markers (*Juczewski et al., 2020*), which could impair learning and interact with other manipulations. Comparisons between head-fixed and freely moving behaviors should be made cautiously, as the stress produced by head fixation may influence behaviors and neuronal activity. Future studies should investigate the relationship between different head-fixed approaches and stress responses to determine best practices for reducing stress associated with head fixation in mice. The advantage of limiting the range of behaviors a subject can display comes at the cost of reduced naturalistic character, which can impair behavior and related neuronal activity (*Aghajan et al., 2015*; *Aronov and Tank, 2014*). Furthermore, isolation of components of behavior provides powerful insight into the neuronal mechanisms that underlie the particular component of behavior but may impair insight into how the related neuronal circuits function during more complex behaviors and contexts. Even in the presence of these caveats, extensive research conducted in head-fixed non-human primates has made vast progress in a multitude of areas of neuroscience (*Mirenowicz and Schultz, 1996*; *Parker and Newsome, 1998*; *Schultz et al., 1997*) including appetitive and consummatory behaviors (*Bromberg-Martin et al., 2010*; *Haber and Knutson, 2010*). The greatest insights into the neuronal mechanisms of behavior will come from a mixture of both naturalistic behaviors and highly controlled behaviors facilitated by head-fixed behaviors made possible with OHRBETS.

The OHRBETS platform presented here was designed to be scalable, flexible, and compatible with external hardware. By using low-cost, open-source, and 3D printed components and publishing extensive instructions for assembly, our system is affordable and scalable across labs of all sizes and

budgets. Despite the use of low-cost and 3D printed components, our system is remarkably consistent and reliable across hundreds of behavioral sessions. Our hardware and software are modular, as all hardware components can be easily swapped, and all behavioral programs are written to produce data with a uniform format. Using different combinations of components will facilitate conducting a wide variety of behavioral experiments including all the experiments presented in this manuscript and many more. By using an Arduino Mega case as a microprocessor mounted within a 3D printed enclosure, one can integrate many different forms of connectivity to interface with external hardware. In the online models, we have options for communication via BNC, Cat6, and DB25 that can be easily combined to suit the user's needs. Altogether, OHRBETS is a complete platform for diverse behavioral experiments in head-fixed animals that can be easily adapted by the broader scientific community to conduct an even wider range of procedures that are compatible with monitoring and manipulating neural dynamics in vivo.

# Materials and methods

**Key resources table**

| Reagent type (species) or resource | Designation | Source or reference | Identifiers | Additional information |
|---|---|---|---|---|
| Strain, strain background (*Mus musculus*) | *Mus musculus* with name C57BL/6J | https://www.jax.org/strain/000664 | RRID:IMSR_JAX:000664 | |
| Strain, strain background (*Mus musculus*) | *Mus musculus* with name Slc32a1tm2(cre)Lowl (vgat-cre) | https://www.jax.org/strain/016962 | RRID:IMSR_JAX:016962 | |
| Strain, strain background (*Mus musculus*) | *Mus musculus* with name Slc17a6tm2(cre)Lowl (vglut2-cre) | https://www.jax.org/strain/016963 | RRID:IMSR_JAX:016963 | |
| Strain, strain background (AAV5) | AAV5-EF1a-DIO-hChR2(H134R)-eYFP | UNC Vector Core | | lot #: AV4313Z |
| Strain, strain background (AAV5) | AAV5-Ef1a-DIO-mCherry | UNC Vector Core | | lot #: AV4311E |
| Strain, strain background (AAV9) | AAV9-hSyn-GRAB-DA1h | https://www.addgene.org/113050/ | Catalog #: 113050-AAV9 | lot #: v119464 |
| Strain, strain background (AAV9) | AAV9-hSyn-GRAB-DA2m | https://www.addgene.org/140553/ | Catalog #: 140553-AAV9 | lot #: v140392 |
| Software, algorithm | Sublime Text 3 | https://www.sublimetext.com/3 | | |
| Software, algorithm | Python 3.7 (Anaconda Distribution) | https://www.anaconda.com/ | | |
| Software, algorithm | R 4.0.4 | https://cran.r-project.org/ | | |
| Software, algorithm | RStudio 2022.02.3 build 492 | https://posit.co/download/rstudio-desktop/ | | |
| Software, algorithm | Arduino IDE 1.8.13 | https://www.arduino.cc/ | | |
| Software, algorithm | TinkerCad | https://www.tinkercad.com/ | | |
| Software, algorithm | OHRBETS - Analysis v1.2; | https://github.com/agordonfennell/OHRBETS/tree/main/analysis | | Author: Adam Gordon-Fennell; |

*Continued on next page*

*Continued*

| Reagent type (species) or resource | Designation | Source or reference | Identifiers | Additional information |
|---|---|---|---|---|
| Other | OHRBETS - Open-source hardware | https://github.com/agordonfennell/OHRBETS | | 3D printing models and bill of materials |
| Other | Optic fiber - fiber photometry | https://www.doriclenses.com/ | MFC_400/470–0.37_6mm_MF2.5_FLT | See Materials and methods |
| Other | Optic fiber - optogenetics | https://www.rwdstco.com/ | R-FOC-BL200C-39NA | See Materials and methods; item no: 907-03007-00 |

## Instructions for assembling OHRBETS

Detailed part list, 3D models, electronic wiring diagrams, behavioral programs, and instructions for assembling are available publicly on our GitHub repository (https://github.com/agordonfennell/OHRBETS; copy archived at *Gordon-Fennell, 2023* ). Instructions for assembly are also included as supplementary material (*Supplementary file 1* - assembly protocol). The amount of time necessary to assemble the system will vary depending on the skill of the builder but should be able to be accomplished in approximately 8 hr over 2–3 days.

## Hardware

3D printed components designed and available via the web-based cad software TinkerCAD (Autodesk) and printed using a filament printer (Ultimaker S3) using PLA or resin printer (Form3) using Clear Resin. 3D printed components with 0.15 mm layer height require approximately 52 hr of print time. Components can also be ordered in batch through online 3D printing services to reduce printing demand locally. The micropositioner design was based on one created by *Backyard Brains, 2013*, and the retractable spout design was based on one created by an independent designer (*Buehler, 2016*).

All behavioral hardware was controlled using an Arduino Mega 2560 REV3 (Arduino). The timing of events was recorded via serial communication from the Arduino to the computer (PC, running Windows10) by USB. Lick spouts were made by smoothing 23 gauge blunt fill needles using a Dremel with a sanding disk. Liquid delivery was controlled by solenoids (Parker 003-0257-900) gated by the Arduino, using a 24 V transistor. The retractable spout, radial spout, and wheel brake utilized micro servos (Tower Pro SG92R). Licks on each spout were detected individually using a capacitive touch sensor (Adafruit MPR121) attached to each metal spout. Importantly, the baseline capacitance of each sensor was kept to a minimum and touch thresholds were reduced from standard values (see GitHub for detailed instructions). The MPR121 is compatible with optical experiments but may not be compatible with all electrophysiology approaches and may therefore need to be replaced with other approaches for measuring licks. Micropositioners were assembled from 3D printed components, Super Glue (Loctite Super Glue ULTRA Liquid Control), screws, and nuts.

## Hardware validation

We measured the consistency of the retractable spout extension latency and terminal positions using video recording. We recorded 1000 extension/retractions in five separate retractable spouts using a high-speed video camera (Basler, acA800-510um, 200 fps). We then estimated the position of the spout using DeepLabCut (*Mathis et al., 2018*) and analyzed the position of the spout relative to the mean terminal position of the spout over time. We measured the consistency of the wheel brake latency using experimenter and mouse rotation. We recorded 1770 wheel rotations produced by an experimenter and measured the effect of braking using four separate head-fixed systems. We computed the binned rotational velocity by taking the mean instantaneous velocity within 25 ms time bins (*Figure 2E*, *Figure 2—figure supplement 1H*). We also assessed the rotation following brake engagement with all brake events during all operant data included in *Figure 2*.

## Software

All behavioral programs were written in the Arduino language and executed on the Arduino Mega during the behavioral session. The timing of hardware and behavioral events were sent from the Arduino and recorded on a PC computer (Windows 10) via serial communication or through a fiber photometry console via TTL communication. Fiber photometry data was collected using Synapse (Tucker Davis Technologies). Data processing, statistical analysis, and data visualization were performed using custom scripts in Python (version 3.7) and R (version 4.0.4). All behavioral programs and pre-processing scripts used to produce the data in this manuscript are freely available through our GitHub (https://github.com/agordonfennell/OHRBETS) and all visualization and analysis scripts are available through our Zenodo repository (10.5281/zenodo.8015631).

## Animals

This study was performed in strict accordance with the recommendations in the Guide for the Care and Use of Laboratory Animals of the National Institutes of Health. All animal procedures were pre-approved by Animal Care and Use Committees (IACUC) at the University of Washington (#4450-01) or University of Illinois at Chicago (#20-031). A mixture of wild-type and transgenic mice on a C57BL/6J background were used for behavioral experiments throughout the paper. All mice were bred in the lab from mouse lines obtained from Jackson Laboratory aside from 16 wild-type mice obtained directly from Jackson Laboratory. No differences were observed across transgenic lines, so all data was pooled. For optogenetic experiments, $Slc32a1^{Cre}$ (Vgat-Cre) or $Slc17a6^{Cre}$ (Vglut2-Cre) mice (*Vong et al., 2011*) were obtained from Jackson Laboratory and bred in the lab to produce heterozygous offspring used for experiments. Mice used in fiber photometry and optogenetic experiments were singly housed to prevent damage to the optical fibers while all other mice were group-housed. Mice were at least P55 prior to surgery and all groups consistent of both males and females. Mice were assigned to groups randomly at the start of the experiment and the experimenter was not blinded to group identity. Mice were kept on a reverse 12 hr light/dark cycle and behavioral experiments were conducted within the dark cycle unless otherwise noted.

## Surgeries

Mice were anesthetized using isoflurane (5% induction, 1.5–2% maintenance), shaved using electric clippers, injected with analgesic (carprofen, 10 mg/kg, s.c.), and then mounted in a stereotaxic frame (Kopf) with heat support. Skin overlying the skull was injected with a local anesthetic (lidocaine, 2%, s.c.) and then sterilized using ethanol and betadine. Next, an incision was made using a scalpel, and the skull was cleared of tissue and scored using the sharp point of a scalpel. The skull was leveled, two burr holes were drilled in the lateral portion of the occipital bone, and two micro screws were turned into the bone. We then coated the bottom of a stainless steel head ring (custom machined, see GitHub for design) with Super Glue, placed it onto the skull of the mouse, and then encased the head ring and skull screws with dental cement making sure the underside of the ring remained intact. After the dental cement had time to fully dry, the mouse was removed from the stereotaxic frame and allowed to recover with heat support before being returned to their home cage. Mice were allowed to recover for at least 1 week prior to dietary restriction.

Mice used for optogenetic or fiber photometry experiments underwent the same procedure as above with the addition of a viral injection and fiber implantation. Following implantation of skull screws, we drilled a burr hole overlaying the brain region target. We then lowered a glass injection pipette into the target brain region and injected the virus at a rate of 1 nL/s using a Nanoject III (Drummond), waited 5 min for diffusion, and then slowly retracted the pipette. For optogenetic experiments, we injected 300 nL of AAV5-EF1a-DIO-hChR2(H134R)-eYFP (titer: 3.2e12) or AAV5-Ef1a-DIO-mCherry (titer: 3.3e12), and for fiber photometry experiments, we injected 400 nL of AAV9-hSyn-GRAB-DA1h (titer: 2.7e13) or AAV9-hSyn-GRAB-DA2m (titer: 2.4e13). The following stereotaxic coordinates (relative to bregma) were used for injection targets: LHA (0° angle; AP: –1.3 mm; ML: ±1.1 mm; DV: –5.2 mm), NAc medal shell (10° angle; AP: 1.7 mm; ML: ±1.5 mm; DV: –4.8 mm), and NAc lateral shell (10° angle; AP: 1.7 mm; ML: ±2.5 mm; DV: –4.6 mm). Next, we lowered a 200 µm optic fiber for optogenetic experiments (1.25 mm ferrule, 6 mm fiber length, RWD) or a 400 µm optic fiber for fiber photometry experiments (2.5 mm ferrule, 6 mm fiber length, MFC_400/470–0.37_6 mm_MF2.5_FLT, Doric) 0.2 mm dorsal to the injection site and then encased the fiber extending from the brain, metal

ferrule, and head ring with Super Glue and dental cement. Mice were allowed to recover for at least 2 weeks prior to behavior or dietary restriction.

## Behavior

### Habituation to head fixation and free-access lick training

Prior to head-fixed behavior, mice were habituated to the experimenter and head fixation stage over four sessions. In the first session, mice were brought into the behavioral room and allowed to explore the head-fixed apparatus to become acquainted with the sights, smells, and sounds of the behavioral box. On the second session, mice were brought into the behavior room and scruffed twice. In the third session, mice were brought into the behavior room, scruffed twice, and then gently had their rear end and hind paws placed in a 50 mL conical twice. After each habituation session, the mouse was immediately provided food or water depending on their deprivation status. Mice undergoing head-fixed operant conditioning for sucrose or head-fixed multi-spout consumption were habituated to head fixation and trained to lick for sucrose in a single 10 min session. During this session, mice were given free access to water (mice used for quinine and NaCl multi-spout experiments [*Figure 5*]) or 10% sucrose (mice used for all other experiments). Free access was approximated using closed-loop delivery of a pulse of fluid (~1.5 µL) each time the mouse licked the spout. During the training session, mice were head-fixed, and the spout was brought forward to gently touch the mouse's mouth to encourage licking before being moved to be positioned ~2–3 mm in front of the mouse's mouth where it remained throughout the session.

### Head-fixed operant conditioning for sucrose

Retractable spout training consisted of three daily sessions of 60 trials with 5 s access periods separated by 20–40 s inter-trial intervals. During each access period, an auditory tone (5 kHz, 80 dB) was played and five pulses of ~1.5 µL sucrose were delivered with a 200 ms inter pulse interval. We delivered pulses of sucrose to encourage licking in bouts and to minimize the chance that a large droplet of sucrose would fall.

Operant conditioning training consisted of six 30 min sessions of initial training, four sessions of increased fixed ratio, and then five to six sessions of progressive ratio. Throughout operant conditioning, one direction of rotation was assigned as the active direction and the opposite direction was assigned as inactive (counterbalanced across mice). Rotation in the active direction earned sucrose delivery. Each sucrose delivery consisted of wheel brake engagement, followed by spout extension and five pulses of ~1.5 µL of 10% sucrose with an inter-pulse interval of 200 ms. During the 3 s access period, the spout remained extended, a 5 kHz auditory tone was presented, and the brake was left engaged. Rotation in the inactive direction led to wheel brake engagement for the same length of time as the total brake time with active rotation, but the spout did not extend, and sucrose was not delivered. During initial training, the fixed ratio of reward was 1/4 turn in the first session and 1/2 turn during the next five sessions. During increased cost sessions, the fixed ratio was increased to one turn. A total of three mice that underwent initial training were removed from progressive-ratio training, two for not learning the task and one that lost their headcap during behavior. During progressive ratio, the wheel turn cost of reward was increased either semilogarithmic: 0.25, 0.5, 0.81, 1.21, 1.71, 2.3, 3.1, 4.1, etc., approximating (*Richardson and Roberts, 1996*) or linearly by 0.5 rotation (0.5, 1.0, 1.5, 2.0, 2.5, 3.0, etc.) each time a reward was earned. The session duration was 1 hr, or 15 min without earning a reinforcer, whichever comes first.

Reversal training was performed in a naive cohort of mice using an identical procedure to initial operant conditioning training, except in session 6 the direction of the wheel rotation that was reinforced was inverted (right turn reinforced → left turn reinforced). Following reversal, the mice were trained on the task for an additional seven sessions of operant conditioning.

### Optogenetic experiments

Vgat-Cre and Vglut2-Cre mice underwent surgery for experiments with optogenetics outlined above. After at least 4 weeks of recovery, mice were trained on RTPT/WTP and optogenetic reinforcement experiments in series. All mice were trained on RTPT/WTP prior to optogenetic reinforcement, but the order of head-fixed and freely moving versions were counterbalanced across mice.

Freely moving RTPT consisted of one session of habituation and six sessions of RTPT with different stimulation frequencies. During habituation, mice were scruffed, attached to an optic fiber, and allowed to explore the RTPT chamber for 10 min. The RTPT chamber was a two-chamber apparatus (50 × 50 × 25 cm³ black plexiglass) with two identical compartments. Over the next six sessions, mice underwent daily 20 min RTPT sessions with stimulation paired with one of the two compartments. The position of each mouse was tracked in real time using Ethovision (Noldus) and when the mouse's center point was detected in one of the two compartments it triggered continuous laser stimulation (5 ms pulses, ~10 mW power, frequencies: 0, 1, 5, 10, 20, 40 Hz). To prevent associations between stimulation and chambers in the RTPT chamber, the stimulation frequency and compartment paired with laser stimulation were counterbalanced across sessions.

Head-fixed WTP consisted of three sessions of habituation and twelve sessions of WTP with different stimulation frequencies (six sessions without and six sessions with a tone indicating the mouse's position). During habituation, mice were habituated to head fixation as outlined above but without sucrose provided. Over the next 12 sessions, mice underwent daily 20 min WTP sessions with stimulation paired to one half of the wheel. Throughout WTP, mice were head-fixed, an optic fiber was connected and covered using blackout tape, and the start of the session was indicated when the wheel brake was disengaged. At the start of the session, the starting wheel position was set in the unpaired zone adjacent to the paired zone. The wheel rotation was tracked by recording the rotation of the wheel relative to the starting position (64 positions/1 rotation) and when the mouse's position was detected in one of the two zones it triggered continuous laser stimulation (5 ms pulses, ~10 mW power, frequencies: 0, 1, 5, 10, 20, 40 Hz). During sessions 1–6 of WTP, there were no extraneous cues indicating which zone the mouse was located in. During sessions 7–12 of WTP, there were tone cues (5 and 10 kHz) that indicated if the mouse was in the paired or unpaired zones of the wheel. To prevent learned associations between stimulation and zones over multiple sessions, we counterbalanced the following factors across sessions: the stimulation frequency, side of the wheel paired with laser stimulation, and the tone paired with laser stimulation.

Freely moving operant conditioning for optogenetic stimulation with positive reinforcement consisted of one session of habituation, four sessions of optogenetic reinforcement training, and six sessions of optogenetic reinforcement with different stimulation frequencies. During habituation, mice were scruffed, attached to an optic fiber, and allowed to explore the optogenetic reinforcement chamber (MED Associates) for 20 min. The optogenetic reinforcement chamber contained two nose pokes with a light cue located inside and a light cue located above each nose poke, as well as an auditory tone generator. Time stamps of hardware and behavioral events were recorded using MED Associates. During daily 20 min optogenetic reinforcement sessions, one nose poke into the active nose poke triggered 1 s of laser stimulation and concurrent illumination of the active nose poke light cues and 5 kHz auditory tone. Nose pokes during the 1 s stimulation period were recorded but did not result in an additional stimulation. Nose pokes in the inactive hole were recorded but had no programmed consequence. To train mice to respond for laser stimulation, mice were run through five sessions of optogenetic reinforcement training with 20 Hz stimulation. To measure the operant response rates across stimulation frequencies, mice were run though an additional six sessions of optogenetic reinforcement with different stimulation frequencies (5 ms pulses, ~10 mW power, frequencies: 0, 1, 5, 10, 20, 40 Hz).

Head-fixed operant conditioning for optogenetic stimulation with positive reinforcement consisted of five sessions of optogenetic reinforcement training and six sessions of optogenetic reinforcement with different stimulation frequencies. During daily 20 min optogenetic reinforcement sessions, wheel rotation in the active direction triggered 1 s of laser stimulation and concurrent 5 kHz auditory tone. The wheel brake was disengaged throughout the behavioral session. Rotation during the 1 s stimulation period was recorded but did not count toward additional stimulation. Rotation in the inactive direction was recorded but had no programmed consequence. Mice were trained over five sessions of optogenetic reinforcement for 20 Hz stimulation with a fixed ratio of 1/4 turn on session 1 and fixed ratio of 1/2 turn on sessions 2–5. To measure the operant response rates across stimulation frequencies, mice were tested over an additional six sessions of optogenetic reinforcement with different stimulation frequencies (5 ms pulses, ~10 mW power, frequencies: 0, 1, 5, 10, 20, 40 Hz) and a fixed ratio of 1/2 turn.

Freely moving operant conditioning for optogenetic stimulation with negative reinforcement consisted of one session of habituation and three sessions of optogenetic reinforcement training. The habituation and behavioral hardware were identical to the freely moving operant conditioning for optogenetic stimulation with positive reinforcement described above. During daily 20 min optogenetic reinforcement sessions, continuous stimulation was turned on at the start of the session and one nose poke into the active nose poke paused laser stimulation for 3 s and triggered concurrent illumination of the active nose poke light cues and 5 kHz auditory tone. Nose pokes during the 3 s pause period were recorded but did not result in an additional pause. Nose pokes in the inactive hole were recorded but had no programmed consequence.

Head-fixed operant conditioning for optogenetic stimulation with negative reinforcement consisted of 11 sessions of optogenetic reinforcement training. During daily 20 min optogenetic reinforcement sessions, laser stimulation was turned on at the start of the session and wheel rotation in the active direction paused laser stimulation for 3 s and triggered a concurrent 5 kHz auditory tone. The wheel brake was disengaged throughout the behavioral session. Rotation during the 3 s pause period was recorded but did not count toward additional pause. Rotation in the inactive direction was recorded but had no programmed consequence. Mice were run through five sessions of optogenetic reinforcement training for 5 Hz stimulation and then six sessions for 10 Hz stimulation with a fixed ratio of 1/4 turn in session 1 and fixed ratio of 1/2 in sessions 2–11.

## Head-fixed multi-spout consumption

Mice were habituated to head fixation as outlined above, and then ran through one to three sessions of spout training (see *Habituation to head fixation and free-access lick training*). The multi-spout assay consisted of daily sessions with 100 trials of 3 s access to one of five different solutions (pseudo-random order with two presentations of each solution per every 10 trials), each session with an inter-trial interval of 5–10 s sampled from a uniform distribution. Licks were detected using a capacitive touch sensor and triggered solution delivery via solenoid opening. The duration of opening for each solenoid was calibrated before each experiment to deliver approximately 1.5 µL per solenoid opening by weighing the weight of water produced with 100 solenoid openings. To control for a spout effect, the pairing of solutions to spouts was counterbalanced such that each spout was paired with each solution over every five sessions. Mice were trained for a minimum of three sessions prior to the five consecutive sessions that are averaged together and used for analysis. Mice in experiments with quinine or NaCl were initially trained in the multi-spout assay with water on all five spouts for three sessions prior to introducing quinine or NaCl solutions. Mice in quinine experiment were tested with the low quinine dilution set (1 mM, 1:4 serial dilution) for eight sessions, high (10 mM, 1:4 serial dilution) for three sessions, and med (5 mM, 1:4 serial dilution) for three sessions in series.

## Homeostatic demand multi-spout experiments

For experiments with alterations in the homeostatic demand for sucrose solution, mice were trained on the multi-spout assay under three homeostatic states in series (water-restricted, food-restricted, then ad libitum). First, mice under water restriction were trained in the multi-spout assay for different concentrations of sucrose over eight sessions. Mice were then removed from water restriction and maintained on food restriction for 1 week prior to being run through the multi-spout assay for five sessions. Finally, mice were removed from all restrictions and maintained with ad libitum access to food and water for three sessions prior to being run through the multi-spout assay for eight sessions. The final five sessions from each homeostatic demand state were used for analysis. Mice that went through the fiber photometry recording experiment were run through the same procedure except the order of water restriction and food restriction was counterbalanced across mice, and they were run for three sessions of free-access spout training.

For experiments with alterations in the homeostatic demand for sodium chloride, mice were trained under water restriction and were then run under two homeostatic states in counterbalanced order (sodium-deplete, sodium-replete). First, mice under water restriction were trained in the multi-spout assay with different concentrations of sodium chloride over 10 sessions. Mice were removed from water restriction and given ad libitum access to water for 48 hr prior to manipulations of sodium demand. To generate sodium demand, we used two injections of diuretic furosemide (50 mg/kg) over 2 days (*Jarvie and Palmiter, 2017*). Mice were weighed, injected with furosemide, and then

placed into a clean cage with bedding for 2 hr before being weighed again to confirm diuretic effect (~5% weight loss). Mice were then returned to a clean home cage with ad libitum access water and sodium free chow (Envigo, TD.90228) (sodium-deplete) or a novel sodium-balanced chow (Envigo, TD.90229) (sodium-replete). Mice underwent the same procedure a second time 24 hr later and then were tested for behavior after an additional 24 hr. Mice were tested in the multi-spout assay under either sodium-deplete or sodium-replete states in a single session. Following 48 hr of ad libitum access to water and standard laboratory chow, mice went through the furosemide treatment and behavioral testing again with the opposite homeostatic state.

## Fiber photometry

Wild-type mice underwent surgery for expression of dopamine sensors and fiber implantation as outlined above (see *Surgeries*) before undergoing multi-spout consumption of sucrose under different homeostatic demand states (see *Homeostatic demand multi-spout experiments*). We recorded dopamine dynamics in the NAc medial shell and lateral shell simultaneously during behavior in the multi-spout assay over three consecutive sessions in each homeostatic demand state. After head fixation, we connected to the mouse's fiber implant, patch cables (Doric, 400 µm, 0.37 NA, 2.5 mm stainless steel ferrules) coupled to a six-port mini cube (Doric, FMC6_IE(400-410)_E1(460–490)_F1(500–540)_E2(555–570)_F2(580–680)_S) that was coupled to an integrated fiber photometry system (Tucker-Davis Technologies, RZ10X). We delivered 405 nm and 465 nm light sinusoidal modulated at 211 Hz and 331 Hz, respectively. The average power for each wavelength was calibrated to 30 µW using a power meter (Lux integrated with the RZ10x) prior to the experiment. The fluorescent emission produced by 405 nm and 465 nm excitation were collected using the same fiber used to deliver light and were measured on a photodetector (Lux) and demodulated during recording. The timing of behavioral events were recorded via TTL communication to the fiber photometry system.

Fiber photometry was analyzed using custom Python and R scripts that are freely accessible through our corresponding public repository (https://zenodo.org/record/8015631). A custom Python script was used to convert raw data into tidy format and then an assortment of custom R functions were used to process the fiber photometry signals. The decay in signal throughout the session for the 405 and 465 channels were corrected by fitting and subtracting a third-degree polynomial to each raw signal. We then normalized the signals by computing z-scores using the mean and standard deviation of the entire session. Using the onset of each access period, we created perievent time histograms with time relative to access onset and then resampled signals to 20 samples per second. We used the 405 signal to assess movement artifacts but did not observe any abrupt changes in fluorescence that typically indicate such artifacts (*Figure 8—figure supplement 5*). The 405 signal was not used to correct the 465 signal because we observed simultaneous opposing signals in the 405 and 465 signal that may be attributed to the fact that 405 is not an ideal isosbestic signal for GRAB-DA. The 465 signal in perievent time histograms was shifted based on the mean signal during the 3 s prior to the onset of the access period. To summarize the dopamine signal during access to each solution, we computed the average and peak signal during the access period.

## Histology

We conducted post hoc histology to determine the location of viral expression and optical fiber locations for mice in optogenetic and fiber photometry experiments. At the conclusion of the experiment, mice were deeply anesthetized and transcardially perfused with 20 mL 1× PBS and 20 mL 4% paraformaldehyde (PFA). Heads were removed and post-fixed in 4% PFA for 24 hr, brains were removed and post-fixed for an additional 24 hr, and then brains were transferred to 30% sucrose until they sank. Brains were frozen at –20°C and sectioned at 40 µm on a cryostat (Leica). Every other brain section was collected in 1× PBS and then mounted on a glass slide. Slides were cover-slipped using the mounting medium Fluoroshield with DAPI for visualizing cell nuclei. Sections that contained the bottom of the optic fiber were imaged with epifluorescence at ×5 magnification (Zeiss, ApoTome2; Zen Blue Edition). The location of optic fibers was determined by mapping the position of the fiber using a mouse brain histology atlas (*Paxinos and Franklin, 2001*). The position of fibers was overlaid onto a vector image of the corresponding atlas section using Illustrator (Adobe).

## Statistical analysis and visualization

All raw data and analysis scripts utilized for this manuscript are freely accessible through our corresponding public repository (10.5281/zenodo.8015631). Details for all statistical results presented in the paper can be found in the stats table included with this manuscript (*Source data 1*). Data with repeated measures design were analyzed using a repeated measure analysis of variance (RM ANOVA) using the afex package (0.28.1) in R. We computed post hoc comparisons using Tukey's honest significant difference (HSD) using the emmeans package (1.6.0) in R. Results with two variables were analyzed using a two-sided unpaired or paired t-tests using base R. Correlations were computed using Pearson's product-moment correlation coefficient using base R. For all statistics, significance was set at p values less than 0.05.

Data was visualized using the ggplot2 (3.3.3) package in R. Unless otherwise noted, error bars depict standard error of the mean and box plots depict median and intraquartile range. Combined plots were assembled using patchwork (1.1.1). Color scales were produced using pals (1.7) or viridis (0.6.2). 3D renderings of the head-fixed hardware were produced using Fusion 360 (Autodesk). Plot components were assembled and further edited using Illustrator (Adobe).

## Acknowledgements

AGF was supported by F32 DA054719, T32 DA7278-27, R01 DA038168, R21 DA050868, and P30 DA048736. GDS was supported by R37 DA032750, R01 DA038168, R21 DA050868, and P30 DA048736. MRB and RG were supported by R37 DA03339, P50MH119467 (project 4), and P30 DA048736. MFR and PB were supported by R21 DA050868.

We would like to thank Dr. Vijay Namboodiri, Dr. Ivan Trujillo-Pisanty, and Madelyn Hjort for providing initial training to AGF in the Arduino language and head-fixed approaches. Lydia Gordon-Fennell for creating vector illustrations of head-fixed mice in figure diagrams and editing the manuscript. Barbara Benowitz for editing the manuscript. Dr. Spencer Smith for providing initial designs for metal head rings and head fixation stage. Dr. Nick Steinmetz for providing insights into the behavior of mice during head-fixed behavior with wheel turning as an operant response.

## Additional information

### Funding

| Funder | Grant reference number | Author |
|---|---|---|
| National Institute on Drug Abuse | DA032750 | Garret D Stuber |
| National Institute on Drug Abuse | DA038168 | Garret D Stuber |

The funders had no role in study design, data collection and interpretation, or the decision to submit the work for publication.

### Author contributions

Adam Gordon-Fennell, Conceptualization, Resources, Data curation, Software, Formal analysis, Supervision, Validation, Investigation, Visualization, Methodology, Writing – original draft, Project administration, Writing – review and editing; Joumana M Barbakh, MacKenzie T Utley, Shreya Singh, Paula Bazzino, Raajaram Gowrishankar, Investigation, Writing – review and editing; Michael R Bruchas, Resources, Supervision, Writing – review and editing; Mitchell F Roitman, Resources, Supervision, Funding acquisition, Project administration; Garret D Stuber, Conceptualization, Resources, Supervision, Funding acquisition, Project administration, Writing – review and editing

### Author ORCIDs

Adam Gordon-Fennell 🔟 http://orcid.org/0000-0002-7550-5084
MacKenzie T Utley 🔟 http://orcid.org/0000-0001-9488-0819
Michael R Bruchas 🔟 http://orcid.org/0000-0003-4713-7816
Garret D Stuber 🔟 http://orcid.org/0000-0003-1730-4855

## Ethics

This study was performed in strict accordance with the recommendations in the Guide for the Care and Use of Laboratory Animals of the National Institutes of Health. All animal procedures were pre-approved by Animal Care and Use Committees (IACUC) at the University of Washington (#4450-01) or University of Illinois at Chicago (#20-031).

## Decision letter and Author response

Decision letter https://doi.org/10.7554/eLife.86183.sa1
Author response https://doi.org/10.7554/eLife.86183.sa2

## Additional files

### Supplementary files

• Supplementary file 1. Assembly protocol. Detailed protocol for OHRBETS (Open-Source Head-fixed Rodent Behavioral Experimental Training System) assembly.

• MDAR checklist

• Gordon-FennellASource data 1. Table of statistical results.

### Data availability

All data and analysis code for the paper to reproduce figures and findings can be found at: 10.5281/zenodo.8015631. Repository of code required to build the system can be found at: https://github.com/agordonfennell/OHRBETS (copy archived at *Gordon-Fennell, 2023*).

The following dataset was generated:

| Author(s) | Year | Dataset title | Dataset URL | Database and Identifier |
|---|---|---|---|---|
| Adam G F, Garret S | 2023 | An Open-Source Platform for Head-Fixed Operant and Consummatory Behavior | https://doi.org/10.5281/zenodo.8015631 | Zenodo, 10.5281/zenodo.8015631 |

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
