## [Editor Report]

This important study by Gordon-Fennell et al. presents a low-cost, open-source platform for measuring action elicitation and consummatory behavior in head-fixed animals. The authors present exceptional evidence that this platform allows animals to perform a truly voluntary action whilst their head is held still. The results have the potential to have a broad impact in the field as many labs start to move towards measuring head-fixed behavior effectively, although this is said with the caveat that such behavior will never be an ideal replication of naturalistic behavior.

---

## [Decision Letter]

**Decision letter after peer review:**

Thank you for submitting your article "An Open-Source Platform for Head-Fixed Operant and Consummatory Behavior" for consideration by *eLife*. Your article has been reviewed by 3 peer reviewers, including Laura A Bradfield as the Reviewing Editor and Reviewer #1, and the evaluation has been overseen by Kate Wassum as the Senior Editor. The following individuals involved in the review of your submission have agreed to reveal their identity: Erin Calipari (Reviewer #2); Ranier Gutiérrez (Reviewer #3).

The reviewers have discussed their reviews with one another, and the Reviewing Editor has drafted this to help you prepare a revised submission. With your revision, please provide a point x point response to reach reviewer critique in the public and private reviews, paying special attention to the essential revisions below.

Essential revisions:

1) Please rectify issues regarding terminology and density to make the manuscript more accessible to readers. In particular, we found the use of the term "negative ICSS" in Figure to be confusing as, although it is negative reinforcement, it is not clear what "negative ICSS" comprises. Please see specific reviewer comments for more details on this point.

2) Each reviewer was concerned about the influence of spatial cues (or lack of influence where one would be expected) on results. For instance, we had concerns regarding the putative replication of real-time place preference tasks given the absence of a spatial element in the head-fixed task. Reviewer #3 also had concerns regarding the spatial locations of each sipper that might be encoded as a cue. Please either add caveats to the interpretation of the current tasks or move the relevant figures to supplemental material.

3) Please include a key resource table.

4) Please ensure your manuscript complies with the *eLife* policies for statistical reporting: https://reviewer.elifesciences.org/author-guide/full "Report summary statistics (e.g., t, F values) and degrees of freedom, exact p-values, and 95% confidence intervals wherever possible. These should be reported for all key questions and not only when the p-value is less than 0.05." This should be in the main manuscript.

*Reviewer #1 (Recommendations for the authors):*

1. As I do not do head-fixed research, I would just like to know how novel this system is if the authors could please comment.

2. I suggest the authors include more detail in the manuscript on how to build the system.

3. I suggest the authors include a statement as to their purpose for comparing across experimental set-ups (i.e. to show the similarities in systems despite the alterations in variables).

4. Personally I would remove Figure 3 from the manuscript altogether. However, if it is kept in, the authors definitely need to add some caveats discussing how the head-fixed version differs from the free-moving version, as well as noting that the brain cells/circuits involved may differ between tasks.

5. My preference would be to have the statistics reported in the paper so these can be easily accessed alongside the data they are referring to.

6. Perhaps authors could clarify their reasoning for making this inference about medial shell without engaging in circular logic.

7. Grammatical and other errors:

a. LHA is never spelled out in full (and so before I looked at the diagrams I thought the authors were referring to the lateral habenula).

b. Line 199 there is an extra 'a' between males and females.

c. Session should be plural in both cases, line 277.

8. I also wasn't sure if NaCl could be called 'strongly aversive as they keep drinking it throughout the session at low concentrations. Perhaps the authors could clarify this about the concentrations.

*Reviewer #2 (Recommendations for the authors):*

I do have some suggestions for the authors that I believe would improve rigor and accessibility.

1) The authors present an impressive collection of experiments and behavioral analysis as validation of the OHRBETS platform, however, some of the extensive analyses and data portrayal may make the straightforward findings relatively difficult to digest as a reader (especially one that does not have a strong background in reinforcement learning). A general streamlining of each figure may improve readability/accessibility. Some suggestions are below:

a. Figure 1: A single graph is likely sufficient to convey each behavioral manipulation (i.e. response cost, progressive ratio, contingency reversal). When presented in bulk, a reader that is not as familiar with different behavioral portrayals may get bogged down in multiple representations of a singular message. Individual session records are informative (and suitable for a supplementary figure) but may be unnecessary here.

b. Alternatively, Figure 1A-E (Apparatus diagram and mechanical validation) may be better served as a stand-alone figure, apart from Figure1F-Q.

c. Figure 2: The correlations in panel I don't provide any information that can't be discerned from the analysis conveyed in panel H.

d. This is true of Figure 3F as well.

e. Analysis such as in Figure 3 Supplement 1C are excessive – p-values, correlation coefficients, etc. should not be parametrically compared and don't need to be visually depicted.

2) It's not clear that the wheel-turning procedure depicted in Figure 3 is entirely analogous to a place preference procedure. RTPP procedures in freely-moving animals likely involve spatial learning/associative processes that may not be necessary to perform the wheel-turning task (under the control of an auditory cue). A task involving visual VR (perhaps with tactile stimuli discriminating wheel position) may be better suited for an RTPP analogue. This could be addressed by not specifically describing this task in this way.

3) "Negative ICSS" is not the appropriate term for the task described in figure 2, as the animals avoid/escape stimulation as opposed to 'self-stimulating'. Revise the language here for clarity – e.g positive and negative reinforcement with ICS, ICS avoidance, etc.

4) The circadian effects on consummatory behavior are orderly but don't provide critical validation of consummatory measures – while in line with the literature, this may be more appropriate as a supplemental figure; however, I will leave that up to the authors to decide what they think is appropriate.

5) Repeatedly referring to the platform as an 'ecosystem' more confusing than descriptive and limiting jargon would make the manuscript more accessible.

*Reviewer #3 (Recommendations for the authors):*

The manuscript is quite relevant to the Neuroscience field and will be of general interest to the *eLife* audience. The experiments were carefully done. It is expected that OHRBETS will be widely used in multiple Neuroscience labs. However, I have some major comments that the authors need to address to improve the manuscript.

The authors failed to discuss the major caveat of OHRBETS, since each sipper is spatially put in a different location (rather than putting them in a bundle, see https://doi.org/10.1152/ajpregu.00492.2012). Hence, with overtraining, this spatial exteroceptive information could be used by mice to anticipate taste identity. For example, if one concentration is Quinine and the other different tastants (sucrose, NaCl, HCl), is it possible that mice learn to predict Quinine or sucrose stimuli? This experiment is not shown since the Authors used different mice and only tested arrays of concentrations of one taste quality. I would like the authors to discuss the possibility that mice could use the spatial location of sippers as a cue. Perhaps a future device could be improved by using needles in a bundle rather than a circular configuration of the sippers.

The introduction is overly optimistic and biased towards only the positive benefits of head-fixed preparations. It will benefit to briefly introduce some caveats of the technique right from the introduction, for example, authors wrote:

"Eliminating or controlling physical approach behaviors also allows for isolation of both appetitive and consummatory behaviors and related neuronal dynamics. By removing turning associated with locomotion, head-fixed behavioral approaches offer enhanced compatibility with neuroscience approaches that require tethers including optogenetics and fiber-photometry".

It is a common misconception to assume that head-fixed mice are not moving at all, they, in fact, continue performing subtle movements during head fixation, we can measure them using cell loads. For a beautiful demonstration of this, see and cite Hughes and Henry Yin's 2020 work https://www.frontiersin.org/articles/10.3389/fnint.2020.00011/full

On page 17 "Furthermore, these data indicate that OHRBETS is highly compatible with neural recording and manipulation techniques that would be challenging with freely moving behavioral designs."

On page 19 Discussion. "Eliminating locomotion improves compatibility with many standard neuroscience approaches including optogenetics, fiber-photometry, electrophysiology, and calcium imaging."

It is still possible that the use of capacitive touch sensors could introduce an electrical artifact. Thus, it is unknown if this device is really compatible with electrophysiological recordings, such as neuropixels.

More importantly, although the authors mention it briefly in the discussion, the Stress induced by restricting the mice is an enormous difference between freely moving and head-fixed preparations that should always be highlighted. The authors did not measure stress hormones, but some previous work shows this effect.

At the discussion, nothing is mentioned about NAcShL dopamine results!! DA release in this region was more robust than NAcShM.

---

## [Author Response]

Essential revisions:1) Please rectify issues regarding terminology and density to make the manuscript more accessible to readers. In particular, we found the use of the term "negative ICSS" in Figure to be confusing as, although it is negative reinforcement, it is not clear what "negative ICSS" comprises. Please see specific reviewer comments for more details on this point.

We agree that the phrase “Negative ICSS” may be confusing. We have revised the language to refer to “Negative ICSS” as negative optogenetic reinforcement and “Positive ICSS” as positive optogenetic reinforcement.

2) Each reviewer was concerned about the influence of spatial cues (or lack of influence where one would be expected) on results. For instance, we had concerns regarding the putative replication of real-time place preference tasks given the absence of a spatial element in the head-fixed task. Reviewer #3 also had concerns regarding the spatial locations of each sipper that might be encoded as a cue. Please either add caveats to the interpretation of the current tasks or move the relevant figures to supplemental material.

We thank the reviewers for raising these concerns.

To avoid confusion regarding the place component of the real-time place preference assay name, we have renamed the head-fixed assay for assessing valence to Wheel-Time Preference (WTP). We have also added a full paragraph to the discussion where we outline the differences in the task requirements and relevant neuronal circuits between the freely-moving RTPP and head-fixed WTP. We understand that the head-fixed task is not a perfect analog of the RTPP task, however based on the similarity in the resulting time spent in the stimulation chamber/zone we believe that the WTP is able to replicate the valence assessment that many in the field uses RTPP to measure. We believe that the WTP with OHRBETS opens up new possibilities for assessing preference in head-fixed mice and this justifies keeping the figure within the main manuscript.

To thoroughly address the potential confound of spatial information during the multi-spout experiment, we have added an additional supplemental figure (Figure 4—figure supplement 5) that depicts the proportion of trials with licking and added a paragraph to the discussion centered on the potential confound associated with learning the solution identity. Please see a more thorough outline of the changes made in the Reviewer 3 point by point response.

3) Please include a key resource table.

We have included a key resource table with this submission.

4) Please ensure your manuscript complies with the eLife policies for statistical reporting: https://reviewer.elifesciences.org/author-guide/full "Report summary statistics (e.g., t, F values) and degrees of freedom, exact p-values, and 95% confidence intervals wherever possible. These should be reported for all key questions and not only when the p-value is less than 0.05." This should be in the main manuscript.

Thank you for raising this concern. With our initial submission, we were concerned that including all of the statistics within the main text would make the paper difficult to read due to the extensive amount of statistics. With this submission, in addition to the statistics table, we have included statistics within the figure legends and main text where applicable.

Reviewer #1 (Recommendations for the authors):1. As I do not do head-fixed research, I would just like to know how novel this system is if the authors could please comment.

The novelty of the system stems from the synergistic combination of functionality, the low-cost open source nature of the design, and the breadth of behavioral procedures the system is able to support. The use of a wheel as an operant response was adapted from the International Brain Laboratory rig which has been used extensively for visual discrimination tasks. We adapted this wheel design to make the response closer to lever pressing through the use of the wheel brake, which ensures that subjects have to rotate the wheel in discrete rotational bouts rather than continuously spinning the wheel and potentially disengaging and allowing the wheel to rotate independently. There are no examples of systems capable of delivering 5+ solutions within a behavioral session or conducting valence testing with a modification of real-time place preference without the cost and complexity associated with virtual reality. We believe that the combination of factors, the flexibility and scalability of the system makes OHRBETS a novel and useful system for diverse motivation and consumption behaviors in head-fixed mice.

2. I suggest the authors include more detail in the manuscript on how to build the system.

With this submission we have included detailed assembly instructions as a supplement to the main manuscript and added reference to the file within the methods section. We have also added details, including time estimates, to the methods section.

3. I suggest the authors include a statement as to their purpose for comparing across experimental set-ups (i.e. to show the similarities in systems despite the alterations in variables).

Thank you for highlighting this. We have added in a justification for why we measured the consistency in behavior measured with each head-fixed system.

4. Personally I would remove Figure 3 from the manuscript altogether. However, if it is kept in, the authors definitely need to add some caveats discussing how the head-fixed version differs from the free-moving version, as well as noting that the brain cells/circuits involved may differ between tasks.

Please see our response in Essential Revisions.

5. My preference would be to have the statistics reported in the paper so these can be easily accessed alongside the data they are referring to.

Please see our response in Essential Revisions.

6. Perhaps authors could clarify their reasoning for making this inference about medial shell without engaging in circular logic.

Please see our response in Essential Revisions.

7. Grammatical and other errors:a. LHA is never spelled out in full (and so before I looked at the diagrams I thought the authors were referring to the lateral habenula).b. Line 199 there is an extra 'a' between males and females.c. Session should be plural in both cases, line 277.

These problems have been addressed in this resubmission.

8. I also wasn't sure if NaCl could be called 'strongly aversive as they keep drinking it throughout the session at low concentrations. Perhaps the authors could clarify this about the concentrations.

Thank you for highlighting this potential miscommunication. We have edited the text to specifically reference that the strongly aversive events are in response to the highest concentrations of NaCl.

Reviewer #2 (Recommendations for the authors):I do have some suggestions for the authors that I believe would improve rigor and accessibility.1) The authors present an impressive collection of experiments and behavioral analysis as validation of the OHRBETS platform, however, some of the extensive analyses and data portrayal may make the straightforward findings relatively difficult to digest as a reader (especially one that does not have a strong background in reinforcement learning). A general streamlining of each figure may improve readability/accessibility. Some suggestions are below:a. Figure 1: A single graph is likely sufficient to convey each behavioral manipulation (i.e. response cost, progressive ratio, contingency reversal). When presented in bulk, a reader that is not as familiar with different behavioral portrayals may get bogged down in multiple representations of a singular message. Individual session records are informative (and suitable for a supplementary figure) but may be unnecessary here.

Thank you for this suggestion. We have trimmed down multiple figures to address this concern. We have kept the behavioral data presentation in figure 1 because we believe that including both single subject data and group means / error estimates provides the most transparent and thorough representation of the data.

b. Alternatively, Figure 1A-E (Apparatus diagram and mechanical validation) may be better served as a stand-alone figure, apart from Figure1F-Q.

Thank you for this suggestion, we agree that the information conveyed in figure 1 would be better suited as two separate figures, one for the system design and one for the behavior. We have broken up the figure and added additional system information to the new system design figure.

c. Figure 2: The correlations in panel I don't provide any information that can't be discerned from the analysis conveyed in panel H.d. This is true of Figure 3F as well.

While we agree that the information presented in the scatter plots can be derived from the line plots, we believe that the scatter plots do a more effective job of precisely conveying the relationship between the head-fixed and freely-moving behavior over the range of the behavioral phenotype. However, we also agree that the information may overload our audience, so we have removed the scatter plots from the main figures and moved them to supplemental figures for those who want to see the data in greater detail.

e. Analysis such as in Figure 3 Supplement 1C are excessive – p-values, correlation coefficients, etc. should not be parametrically compared and don't need to be visually depicted.

Thank you for providing this suggestion. Figure 3—figure supplement 1c and the related text in the Results section have been removed.

2) It's not clear that the wheel-turning procedure depicted in Figure 3 is entirely analogous to a place preference procedure. RTPP procedures in freely-moving animals likely involve spatial learning/associative processes that may not be necessary to perform the wheel-turning task (under the control of an auditory cue). A task involving visual VR (perhaps with tactile stimuli discriminating wheel position) may be better suited for an RTPP analogue. This could be addressed by not specifically describing this task in this way.

Please see our response in Essential Revisions.

3) "Negative ICSS" is not the appropriate term for the task described in figure 2, as the animals avoid/escape stimulation as opposed to 'self-stimulating'. Revise the language here for clarity – e.g positive and negative reinforcement with ICS, ICS avoidance, etc.

We agree that the phrase “Negative ICSS” may be confusing. We have revised the language to refer to “Negative ICSS” and negative optogenetic reinforcement and “Positive ICSS” as positive optogenetic reinforcement.

4) The circadian effects on consummatory behavior are orderly but don't provide critical validation of consummatory measures – while in line with the literature, this may be more appropriate as a supplemental figure; however, I will leave that up to the authors to decide what they think is appropriate.

While we agree with the reviewer that the findings are not a critical validation of the system, we included the results as a figure in the paper because of the low level of licking observed during the initial free-access consumption training sessions. We believe that including this figure in the main text will help users of the system replicate and expand upon the multi-spout results presented elsewhere in the paper.

5) Repeatedly referring to the platform as an 'ecosystem' more confusing than descriptive and limiting jargon would make the manuscript more accessible.

We used the word ecosystem to try and communicate that OHRBETS was intended to be a living system that will grow with additional hardware and software moving forward. However, we agree that the word could be confusing so we have replaced it with “platform”.

Reviewer #3 (Recommendations for the authors):The manuscript is quite relevant to the Neuroscience field and will be of general interest to the eLife audience. The experiments were carefully done. It is expected that OHRBETS will be widely used in multiple Neuroscience labs. However, I have some major comments that the authors need to address to improve the manuscript.The authors failed to discuss the major caveat of OHRBETS, since each sipper is spatially put in a different location (rather than putting them in a bundle, see https://doi.org/10.1152/ajpregu.00492.2012). Hence, with overtraining, this spatial exteroceptive information could be used by mice to anticipate taste identity. For example, if one concentration is Quinine and the other different tastants (sucrose, NaCl, HCl), is it possible that mice learn to predict Quinine or sucrose stimuli? This experiment is not shown since the Authors used different mice and only tested arrays of concentrations of one taste quality. I would like the authors to discuss the possibility that mice could use the spatial location of sippers as a cue. Perhaps a future device could be improved by using needles in a bundle rather than a circular configuration of the sippers.

Thank you for this feedback. The task was designed to reduce any predictive information from the subject calibrating rotational position and linear position of the multi-head such that the terminal position of each spout was located in roughly the same spatial position relative to the subject. We also counterbalance the spout ID and the solution over sessions to minimize over training of the subjects. However, despite these mitigating steps, the mice show less trials with engaging in licking during the high molar NaCl during the multi-NaCl sessions (see attached figure). This indicates to us that mice are capable of learning to at least partially predict the solution identity prior to tasting the solution while still maintaining enough trials with licking to measure neuronal correlates in response to aversive tastants. The ability to predict solutions prior to licking does not seem to be present during the multi-quinine sessions, possibly due to the substantially lower number of completed trials or a poorer ability to distinguish between difference concentrations. We chose a radial head design in order to eliminate the risk of tastant contamination across trials, but we agree that a needle bungle approach should be effective at reducing the solution prediction and future versions of the system could incorporate this approach.

We have included the data presented here as a supplemental figure and we have included a paragraph discussing the potential confound and possible mitigation within the Discussion section.

The introduction is overly optimistic and biased towards only the positive benefits of head-fixed preparations. It will benefit to briefly introduce some caveats of the technique right from the introduction, for example, authors wrote:

We now introduce the downsides of reduced naturalistic behavior and enhanced stress in the second sentence of the introduction. While we agree that it is important to discuss the downsides up front, we would like to keep the introduction as succinct as possible. We have expanded the limitations in the discussion to communicate the downsides of head-fixation.

"Eliminating or controlling physical approach behaviors also allows for isolation of both appetitive and consummatory behaviors and related neuronal dynamics. By removing turning associated with locomotion, head-fixed behavioral approaches offer enhanced compatibility with neuroscience approaches that require tethers including optogenetics and fiber-photometry".It is a common misconception to assume that head-fixed mice are not moving at all, they, in fact, continue performing subtle movements during head fixation, we can measure them using cell loads. For a beautiful demonstration of this, see and cite Hughes and Henry Yin's 2020 work https://www.frontiersin.org/articles/10.3389/fnint.2020.00011/fullOn page 17 "Furthermore, these data indicate that OHRBETS is highly compatible with neural recording and manipulation techniques that would be challenging with freely moving behavioral designs."On page 19 Discussion. "Eliminating locomotion improves compatibility with many standard neuroscience approaches including optogenetics, fiber-photometry, electrophysiology, and calcium imaging."

We completely agree that there are subtle movements during head-fixation. In the sentences you reference here, we were intending to specifically reference locomotion defined as coordinated patterns of movements that result in bodily movement through space. By head-fixing the mice we prevent them from engaging in locomotion approach behaviors and we eliminate rotational locomotion relative to the experimental equipment. To ensure that we are accurately communicating this, we have updated the manuscript to specify that while we prevent locomotion preceding consumption we are not eliminating all movement of the animal: “Eliminating or controlling physical approach behaviors, but not all movement (Hughes et al., 2020), also allows for greater isolation of both appetitive and consummatory behaviors and related neuronal dynamics”

It is still possible that the use of capacitive touch sensors could introduce an electrical artifact. Thus, it is unknown if this device is really compatible with electrophysiological recordings, such as neuropixels.

The MPR121 is compatible with some electrophysiological approaches for recording single units (for example see Ottenheimer, D.J., Bari, B.A., Sutlief, E. et al. A quantitative reward prediction error signal in the ventral pallidum. Nat Neurosci 23, 1267–1276 (2020). https://doi.org/10.1038/s41593-020-0688-5). However, according to personal communication with Dr. Ottenheimer, the MPR121 capacitive touch sensor may not be compatible with all neuropixels rigs. We now address the possibility of incompatibility in the methods section after referring to the MPR121.

More importantly, although the authors mention it briefly in the discussion, the Stress induced by restricting the mice is an enormous difference between freely moving and head-fixed preparations that should always be highlighted. The authors did not measure stress hormones, but some previous work shows this effect.

Thank you for this comment. We completely agree that the stress associated with head-fixation is a major potential confound that should be highlighted within the manuscript. To thoroughly acknowledge the potential confounds associated with head-fixation stress, we have added the following in the second sentence of the introduction: “At the cost of reduced naturalistic behavior and enhanced stress…”.

To more thoroughly address the concerns associated with head-fixation stress, we have also expanded the discussion of stress within the caveats paragraph in the discussion:

“Comparisons between head-fixed and freely moving behaviors should be made cautiously, as the stress produced by head-fixation may influence behaviors and neuronal activity. Future studies should investigate the relationship between different head-fixed approaches and stress responses to determine best practices for reducing stress associated with head-fixation in mice.”

At the discussion, nothing is mentioned about NAcShL dopamine results!! DA release in this region was more robust than NAcShM.

We tried to keep the discussion concise but should have included a more detailed discussion of the dopamine release in both the NAcShL and NAcShM. We have added the following discussion:

“Dopamine release in the NAcShL shows higher amplitude increases to the most rewarding solutions, especially during consumption of sucrose in both water and food deprived states (Figure 8). While we observed greater GRAB-DA signals in the NAcShL compared to the NAcShM, it is important to consider that the GRAB-DA signal reflects changes in dopamine release rather than absolute levels of dopamine. This means that the greater signals we observe in the NAcShL could be the consequence of greater dopamine release, or a lower dopamine tone at baseline.”